# Conceptualising surface water-groundwater exchange in braided river systems

Scott R. Wilson[1], Jo Hoyle[2], Richard Measures[2], Antoine Di Ciacca[1], Leanne K. Morgan[3], Eddie W. Banks[4], Linda Robb[1], and Thomas Wöhling [1,5]

[1]Environmental Research, Lincoln Agritech, Lincoln University, New Zealand
[2]National Institute of Water & Atmospheric Research, Christchurch, New Zealand
[3]Waterways Centre for Freshwater Management, University of Canterbury, Christchurch, New Zealand
[4]National Centre for Groundwater Research & Training & College of Science & Engineering, Flinders University, Australia
[5]Chair of Hydrology, Technische Universität Dresden, Dresden, Germany

*Correspondence to*: Scott R. Wilson (scott.wilson@lincolnagritech.co.nz)

**Abstract**

Braided rivers can provide substantial recharge to regional aquifers, with flow exchange

between surface water and groundwater occurring at a range of spatial and temporal scales.

However, the difficulty of measuring and modelling these complex and dynamic river

systems has hampered process understanding and the upscaling necessary to quantify these

fluxes. This is due to an incomplete understanding of the hydrogeological structures which

control river-groundwater exchange. In this paper we present a new conceptualisation of

subsurface processes in braided rivers based on observations of the main losing reaches of

three braided rivers in New Zealand.

The conceptual model is based on a range of data including lidar, bathymetry, coring,

particle size distribution, groundwater, and temperature monitoring, radon-222, electrical

resistivity tomography, and fibre optic cables. The combined results indicate that sediments

within the recently active river braidplain are distinctive, with sediments that are poorly

consolidated and better sorted compared to adjacent deposits from the historical braidplain

which become successively consolidated and intermixed with flood silt deposits due to

overbank flow.

A distinct sedimentary unconformity, combined with the presence of geomorphologically
distinct lateral boundaries, suggests that a "braidplain aquifer" forms within the active river
braidplain through the process of sediment mobilisation during flood events.
This braidplain aquifer concept introduces a shallow storage reservoir to the river system,
which is distinct from the regional aquifer system, and mediates the exchange of flow
between individual river channels and the regional aquifer. The implication of the new
concept is that surface water-groundwater exchange occurs at two spatial scales. The first is
hyporheic and parafluvial exchange between the river and braidplain aquifer. The second is
exchange between the braidplain aquifer and regional aquifer system. Exchange at both
scales is influenced by the state of hydraulic connection between the respective water
bodies. This conceptualisation acknowledges braided rivers as whole "river systems",
consisting of channels, and gravel aquifer reservoir.
This work has important implications for understanding how changes in river management
(e.g., surface water extraction, bank training and gravel extraction) and morphology may
impact groundwater recharge, and potentially on flow, temperature attenuation, and
ecological resilience during dry conditions.
**1.0 Introduction**
This study is motivated by the need to understand processes and quantify losses from
braided river systems to alluvial aquifers. In New Zealand more than 150 river systems are
thought to have braided reaches (Brower et al., 2024), which provide a substantial
component of recharge to alluvial aquifers. For example, in the case of the Wairau Aquifer,
it has been estimated recharge is sourced almost exclusively from the Wairau River
(Wöhling et al. 2018).
Braided rivers are spatially complex, dynamic hydrologic environments which are not easily
measured using field techniques or represented by numerical models. The determination of
flow exchanges between the river and groundwater in such a hydrologically complex and
morphologically dynamic environment is difficult because of their spatial and temporal flow
complexity, numerous and changing number of channels, dynamic bathymetry, coarse
substrate, and tendency for loss of field installations during high flow events. The
complexity of this challenge demands a collaborative effort that leverages the strengths of a
diverse range of disciplines to develop a comprehensive and holistic understanding of the
problem.
Many authors have studied surface-aquifer exchanges, and several reviews have been
conducted on available techniques for measuring exchanges for alluvial rivers in general
(Kalbus et al. 2006, González-Pinzón et al. 2015, Brunner et al. 2017), ephemeral rivers at
different spatial scales (Banks et al., 2011, Shanafield and Cook 2014), and braided rivers
(Coluccio and Morgan, 2019). Based on these reviews, there is a tendency for previous
studies to focus on quantifying exchange fluxes prior to understanding the hydrological
processes that influence the exchange. Ward and Packman (2019) suggest that despite the
large amount of research on river-groundwater exchanges, an accurate predictive,
transferable understanding of the river corridor is lacking. Indeed, a conceptualisation of
how braided rivers relate to their underlying groundwater systems is lacking (Coluccio and
Morgan, 2019). This has led to ambiguity in what measured exchanges represent, and
difficulty in trying to represent braided rivers within aquifer scale models.
The aim of this paper is to formulate a conceptualisation of braided river exchange
processes at both the local (reach) and sub-catchment (aquifer) scale. This conceptualisation
provides a framework for both providing context for field measurements of flow exchange
to be interpreted, and the potential for representation of local scale processes in sub-
catchment models. Incorporating these local scale processes is vital to predict how changes
in the river system can impact groundwater recharge.
The conceptualisation presented here was developed based on field observations but, for
clarity of explanation, we introduce the conceptual framework first and then present the
supporting evidence. Our working definition for hyporheic exchange is local bed-scale
interaction that occurs within a single channel (e.g., a single riffle), whereas parafluvial
exchange occurs between individual channels at larger scales (across a bar or further). We
also distinguish between a river as a series of wetted channels, and a "river system", which
consists of wetted channels plus subsurface flow through the associated braidplain gravels.
"Braidplain" refers to the lateral extent occupied by the river braids, old bar surfaces and
abandoned channels (Warburton, 1996; Gray et al., 2016; Brower et al., 2023). The extent of
wetted channels and recently reworked bed material (bare gravel) at a given point in time
defines the "active braidplain", which also has potential to shift laterally. Lateral adjustment
of the active braidplain may be limited by hillslope margins, terraces or, in managed rivers,
by rock revetments or artificial stop banks (levees) which are typically protected with
vegetative buffers. We refer to the extent within which our study rivers can currently adjust
as the "contemporary braidplain", acknowledging that in two of our study rivers (Wairau
and Ngaruroro) the contemporary braidplain margins are controlled by engineered flood
defences which have narrowed the natural braidplain such that almost the entire
contemporary braidplain is active.

**2.0 Review of existing concepts**
The prevailing conceptualisation for gravel bed rivers in general consists of a surface
channel with an associated bed with some hydraulic resistance (Schälchli 1992, Wu and
Huang 2000) which exchanges water with an associated fluvial or alluvial aquifer (Stanford
and Ward 1993, Poole and Berman 2001). Within the alluvial aquifer lies a hyporheic zone
which functions as an interface between groundwater and surface waters (Stanford and
Ward 1993, Poole and Berman 2001, Boano et al. 2014). The extent of the hyporheic zone is
defined by its function or process of interest, which can be physical, chemical, biological, or
a combination of functions (Ward 2015). Accordingly, the vertical or lateral boundaries of
the hyporheic zone are transient and flexible, and not easily defined spatially (White 1993,
Boulton et al. 1998, Ward 2015). From a hydrological perspective, the hyporheic zone has
been defined as the extent to which surface water enters the high porosity subsurface
beneath and lateral to a stream and returns to the stream surface farther downstream
(Harvey and Wagner, 2000). Authors have suggested the hyporheic zone as extending from
the upper few centimetres of sediment (Boulton et al. 1998, Sophocleous 2002) to larger
scales ($km^3$) constituting a hyporheic corridor (Stanford and Ward 1993). Valett at al. (1996)
predict the extent of the hyporheic zone to be related to catchment lithology, with
interaction being more extensive in sites with higher alluvial hydraulic conductivity, whereas
Boano et al. (2008) predict the infiltration depth to be related to bedform for any given
hydraulic conductivity.
An additional, ecological, concept is that of a riparian zone (Steiger et al. 2005). This extends
river margins beyond the active channel to include the biosphere supported by and
including recent fluvial landforms and inundated or saturated by bank discharge (Hupp and
Osterkamp 1996). Other authors have considered the presence of a parafluvial zone
situated between the hyporheic zone (beneath the river channel) and riparian zone (Holmes
et al. 1994) which accommodates longer flow paths within the alluvium adjacent to the
stream (Bourke et al. 2014, Cartwright and Hoffmann 2016). Our interpretation of the
parafluvial zone is that it constitutes exchange flow within the alluvial aquifer at spatial and
temporal scales beyond what is considered hyporheic.
From a hydrological perspective of braided rivers, the framework that emerges from the
prevailing concepts is one where river-groundwater exchange occurs within an alluvial
aquifer which conveys both hyporheic and parafluvial flowpaths. The proportion of these
flowpath components theoretically depends on the degree to which there is a net loss or
gain in river channel flow. However, the interpretation of river-groundwater exchanges
becomes challenging in a braided river which comprises multiple channels within an alluvial
aquifer. Harvey and Gooseff (2015), Barthel and Banzhaf (2016) and Ward and Packman
(2019) propose that exchange fluxes be considered at different spatial scales: point, local,
sub-catchment, and regional. Following this approach, exchange within individual river
braids or channels can be considered to occur at the point-scale, and the sum of all braids
within a river reach as a local-scale interaction. While process understanding can be
observed and fluxes quantified at the point and local scale, it is imperative to enable an
upscaling of observed processes so that fluxes can be estimated for at least the sub-
catchment (aquifer) scale.
Previous work on subsurface structure in braided rivers has tended to focus on the role of
bed material heterogeneity. A significant body of literature exists to describe braided river
deposits via morphology (Huber and Huggenberger 2016), sedimentology (Huggenberger
and Regli 2006, Theel et al. 2020), geophysics (Pirot et al. 2019), and modelling approaches
(Pirot et al. 2014; 2015, Brunner et al. 2017, Schilling et al. 2022). To date, no conceptual
model has been posed for how a braided river and its associated braidplain gravels (alluvial
aquifer) relate to those of the underlying regional aquifer. While the structural components
of river-groundwater interaction have been identified by previous authors (e.g. Poole and
Berman 2001, Steiger et al. 2005), the identification of clear spatial boundaries between
structural elements has been missing. From a hydrological perspective of understanding
surface water-groundwater interaction, this creates a problem of where the river system
ends, and where the regional groundwater system begins. The uncertainty related to this
lack of spatial definition transfers to the interpretation of field data, whether a sample is
representative of river channel flow, alluvial aquifer (hyporheic or parafluvial zones), or
regional groundwater.
Consequently, representation of braided rivers in numerical models is problematic since
their complexity is not readily captured by a simple conceptualisation. Water exchanges can
potentially be simulated realistically using a fully coupled hydrological model such as
HydroGeoSphere (Therrien et al. 2010, Brunner and Simmons 2012). However, the data
required to parametrise such a model, and the computational demands of the detailed
mesh required to simulate exchanges in braided rivers make this approach only suitable for
point and local scale studies. Furthermore, braided river morphology is so dynamic that a
new bed morphology would be required following each significant flood event. In recent
years, two approaches to simulate the transitions of dynamic bed morphology and
sediments on river-groundwater exchanges have been tested. The first approach applied
the ensemble Kalman filter and areal imagery to assimilate river bed topography and to
update aquifer hydraulic conductivities in a HydroGeoSphere model for a 2-km section of
the Emme River in Switzerland (Tang et al. 2018). The data assimilation scheme strongly
improved predictions of post-flood hydraulic states of the system. The second approach
proposed a pilot point parametrization scheme where both the aquifer properties (hydraulic
conductivity) and the location of the pilot points are inferred, e.g. from river-bed training
images (Khambhammettu et al. 2020). The corresponding Traveling Pilot points (TRIPS)
scheme could potentially be used to describe the transition between discrete states of river
morphology. To some extent these approaches enable the application of fully integrated 3D
models in dynamic river environments of appropriate scale, although their application in a
larger river system or at a larger scale is untested.
The simple river structure offered by the Streamflow-Routing (SFR, Niswonger and Prudic,
2005) and River (Harbaugh, 2005) packages in MODFLOW represent the river as a flux
boundary condition with vertical flow impedance in the bed expressed by a lumped
parameter termed 'streambed conductance'. Previous authors have shown that the concept
of a streambed resistance concentrating all pressure losses, as implemented in MODFLOW,
is questionable in many cases (Anderson 2005, Rushton 2007, Morel-Seytoux et al. 2018, Di
Ciacca et al. 2019). Moreover, even if such a streambed exists, a major physical issue with a
lumped parameter approach is that streambed conductance values in the field are not
homogenous, but vary spatially (Cardenas and Zlotnik 2003, Zhou et al. 2014, Pryshlak et al.
2015, Laube et al. 2018) and temporally (Levy et al., 2011, Wu et al. 2015). In a braided
river, bed material typically consists of a heterogeneous mixture of cobbles, gravels, and
sands, which can have similar characteristics to alluvial sediments located several metres
beneath the bed. Therefore, the streambed conductance concept seems inappropriate to
represent surface water-groundwater exchange in braided river systems.
Different modelling approaches have been trialled to represent braided rivers in New
Zealand. White et al. (2012) conducted a steady state water balance approach to determine
flow losses for a reach of the Waimakariri River. Exchanges between individual channels and
the adjacent alluvial aquifer were determined via mass balance, although the subsurface
components of the exchange were not explicitly described. An alternative approach by
Wöhling et al. (2018, 2020) simulated dynamic Wairau River exchanges at the sub-
catchment scale using MODFLOW. In this case, the braided nature of the river was not
considered, and the river was represented by the SFR package using a stage-width-flow
relationship derived from a representative channel morphology. While this model fitted
river flux and groundwater level data well, the approach employed a streambed
conductance model, which is difficult to reconcile with the river morphology and bed
sediment seen in the field. A particular drawback of the SFR package is its inability to
represent the hyporheic or parafluvial exchange fluxes observed at the point or local scale.
While the Waimakariri and Wairau modelling studies are relatively comprehensive, in both
cases, an understanding of subsurface structure is missing from the river representation.
This lack of knowledge about the structural controls on subsurface flow in the braided river
environment needs clarification to understand what measured and modelled river-
groundwater exchanges represent. In doing this, a more physically realistic method for
representing braided rivers in numerical surface water-groundwater flow models may be
achieved.

**3.0 Proposed conceptualisation**
A conceptual framework is proposed which captures the key elements of water exchanges
in a braided river system. This conceptualisation builds on the previous work of Fox and
Durnford (2003) and Brunner et al. (2009a and b) and recognises that the hydrological
controls on river-groundwater exchange occur at two distinct interfaces within a braidplain
system. Specifically, these two exchange processes can be summarised as:
1.  River channel $\leftrightarrow$ braidplain aquifer (hyporheic and/or parafluvial exchange)

2. Braidplain aquifer ↔ regional aquifer ("river system" - groundwater exchange)

The first exchange interface is within the active braidplain, and occurs between individual river channels (braids) and the local shallow water-table in the river bed sediments, and occurs at the point or local scale. We term the water stored within these river bed sediments the "braidplain aquifer" (BPA), which facilitates hyporheic and parafluvial flow. For a perennially flowing river, the BPA will retain some degree of saturation throughout the year, although unsaturated conditions may occur in the case of intermittent or ephemeral rivers if there are prolonged periods with no river flow. Individual river braids can be in hydraulic connection (gaining or losing), disconnection, or a transitional state relative to the BPA. At the active braidplain interface, all possible river exchange processes can occur regardless of the hydraulic relationship between the BPA and surrounding regional water table. The second exchange interface occurs at both the local (reach) and sub-catchment (aquifer) scale, between the BPA and regional water table.

Central to this conceptualisation is the presence of a distinct BPA immediately beneath the river surface which facilitates exchange at these two interfaces. A similar term "braided-river aquifer" has been used by previous authors (Pirot et al. 2015), although we have not found an associated definition. The BPA functions as a storage medium to exchange water between the river and regional aquifer in braided river systems. Direct exchange of water between the river and regional aquifer can only occur if the river channel is in direct contact with the regional aquifer (i.e., where the surface water extends to the boundary of the BPA, for example where a channel follows the lateral margin of the BPA, or there is a connection at the base of a deep scour pool).

A key feature of the BPA is its extremely high transmissivity, which is a product of the highly dynamic nature of braided rivers. Bedload transport during flood events causes braids to

form, migrate, and be abandoned; processes that re-work the river bed sediments (Bristow
and Best, 1993; Reinfelds and Nanson, 1993). This reworking process strips most fine
sediment (silts and clays) from the river bed, as the shear stresses during floods are too high
for these size fractions to be deposited except in vegetated areas and backwaters. The
result is a braidplain deposit of high transmissivity lag gravels. High transmissivity combined
with the relatively large bed slope of braided rivers and the presence of multiple braids for
water exchange produces a groundwater flow path which is sub-parallel to the
contemporary braidplain, regardless of the regional groundwater flow direction.
Conceptual diagrams have been drawn for situations where the BPA is hydraulically
disconnected from (Figure 1a) and connected to (Figure 1b) the regional aquifer. The first
case (Figure 1a) consists of three functional lithological layers representing a BPA which is
hydraulically disconnected from an underlying regional aquifer by an unsaturated zone. In
this case, water moves vertically from the BPA to the regional aquifer under a unit hydraulic
gradient. A minimum of three layers is required for an unsaturated zone to develop (Fox and
Durnford, 2003), and there must be a sufficient transmissivity contrast between the
impeding horizon and underlying sediments (Brunner et al., 2009a and b). It should be
noted that braided river deposits are lithologically variable and complex, and in most cases
only two functional layers are required due to stratification in the underlying sediments.

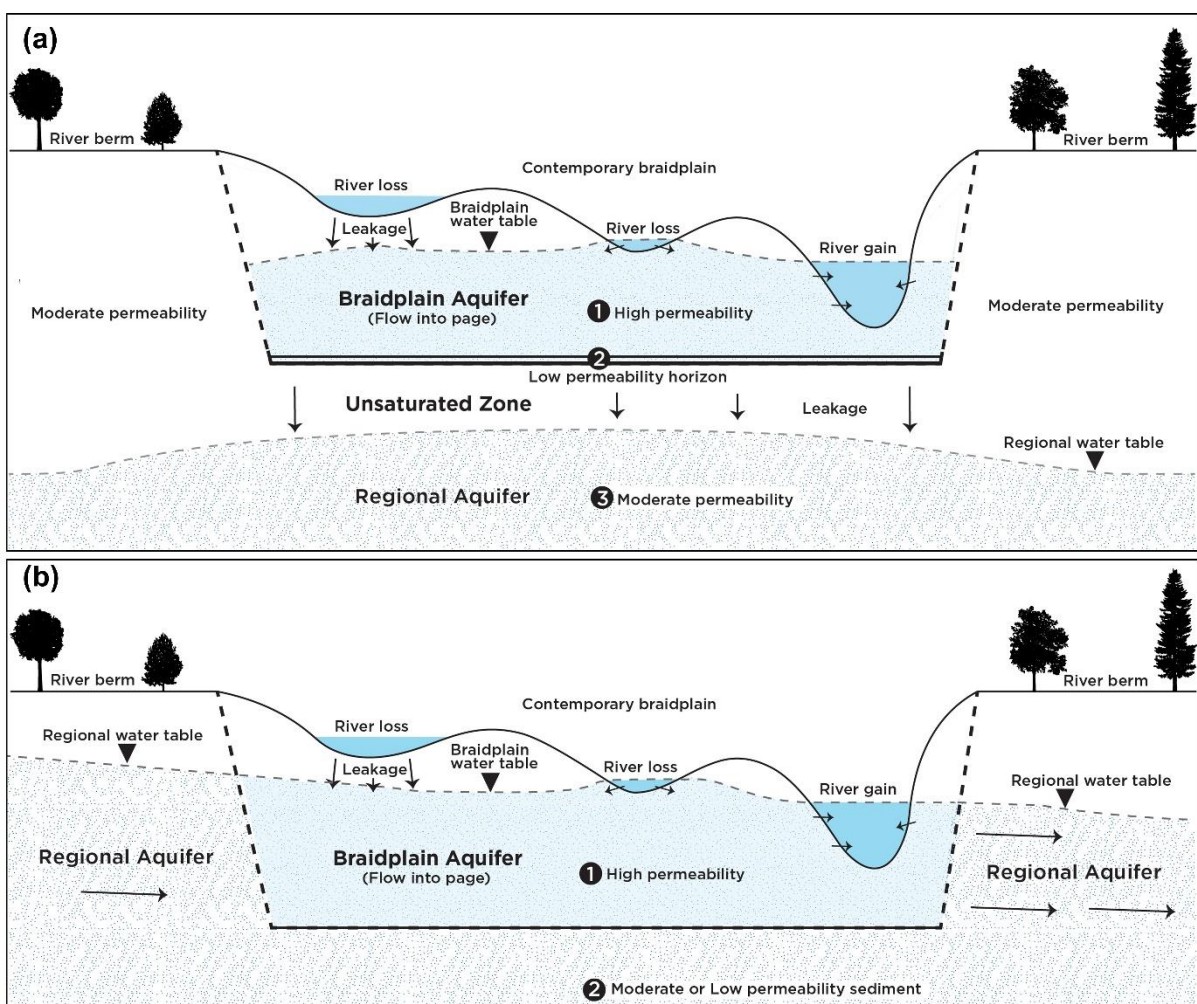

**Figure 1**. Conceptualisation for a braidplain aquifer which is (a) hydraulically disconnected

from or transitional with the regional groundwater system, and (b) hydraulically connected

to the regional groundwater system.

A hydraulically disconnected river system setting shares features in common with

intermittent or ephemeral rivers (Shanafield et al. 2021). Infiltration to the regional aquifer

is regulated by the vertical resistance of lower permeability sediments and the hydraulic

head in the BPA. If the BPA is fully saturated across the contemporary braidplain, the rate of

infiltration to the regional aquifer will be steady because the braidplain has reached a

maximum wetted area. In this condition, some minor temporal infiltration variability will

occur due to changing water levels in the BPA. Under ephemeral conditions, the saturated
extent of the BPA decreases longitudinally, and locally laterally, during drying phases in
response to extended periods of low river flow. The combined reduction in head and
saturated area of wetted braidplain result in considerably less infiltration to the regional
aquifer (Di Ciacca et al. 2023).
The second case is a setting where the river is hydraulically connected to the regional
aquifer system (Figure 1b). A minimum of two lithological units are present, a high
transmissivity BPA and regional aquifer (1) overlying less permeable sediments (2) which
impede vertical flow. The combination of these two factors creates an anisotropy, with
preferential flow in the lateral direction. Hydraulically gaining conditions will occur in
situations where regional water levels are elevated by low permeability boundaries (i.e. the
presence of bedrock) on one or both sides of the river, or at middle to distal positions on
alluvial fans where regional groundwater levels are closer to the land surface.
For a hydraulically connected river system (braids + BPA), the hydraulic gradient is no longer
vertical, as it is for the hydraulically disconnected scenario, but variable, with the exchange
rate governed by the relative hydraulic gradient between the braidplain and regional
aquifers. Thus, groundwater inflow to the BPA or discharge to the regional aquifer can occur
laterally along both margins of the braidplain, as well as vertically through BPA base. The
setting can also be asymmetric, with inflow on one margin, and outflow on the other (as
shown in Figure 1b), with the total river system water balance being gaining or losing, or
having no flow along one margin due to the presence of bedrock.
While the hydraulically connected situation is simpler structurally, relative to the
disconnected situation, exchange between the braidplain and regional aquifers is more
complex. In this case (Figure 1b), water exchange is governed by the hydraulic gradient
between the two aquifers, the transmissivity of both aquifers, and the vertical hydraulic
conductivity of the underlying sediments. Once water exits the BPA in a losing reach, it will
not return unless it is re-routed back to the river system by a reversal of the hydraulic
gradient.
The sedimentological features and groundwater-surface water interaction concepts
associated within the contemporary braidplain have been identified and detailed by
previous authors (e.g., Huggenberger et al. 1998). Regardless of the nature of the
relationship between the braidplain and regional aquifers, the braidplain gravels have a
higher transmissivity than both the adjacent and underlying sediments because of repeated
reworking of the braidplain gravels during high flow events. Hyporheic and parafluvial flow
occurs within these highly transmissive BPA sediments, subparallel to the river flow
direction, with individual braids acting as recharge or discharge boundaries. As such, the
local hydraulic gradient and groundwater flux are largely influenced by river bathymetry.

**4.0 Locations and methods for concept validation**
*4.1 Catchment descriptions*
The conceptualisation presented here is based on field observations of the main losing
reaches of three braided rivers in the drier eastern part of New Zealand (Figure 2). These
are, from north to south, the Ngaruroro (a), Wairau (b), and the Waikirikiri (c, also known as
the Selwyn). These study areas were selected to take advantage of the potential for
hydrological separation afforded by dominantly losing river reaches. The downward
hydraulic gradient, and potential hydrological separation between channel, bed gravels and
regional aquifer enable the possibility for different structural and hydrological components
to be identified. Summary hydrological and geomorphological statistics for the three study
sites and their source catchments are shown in Table 1.

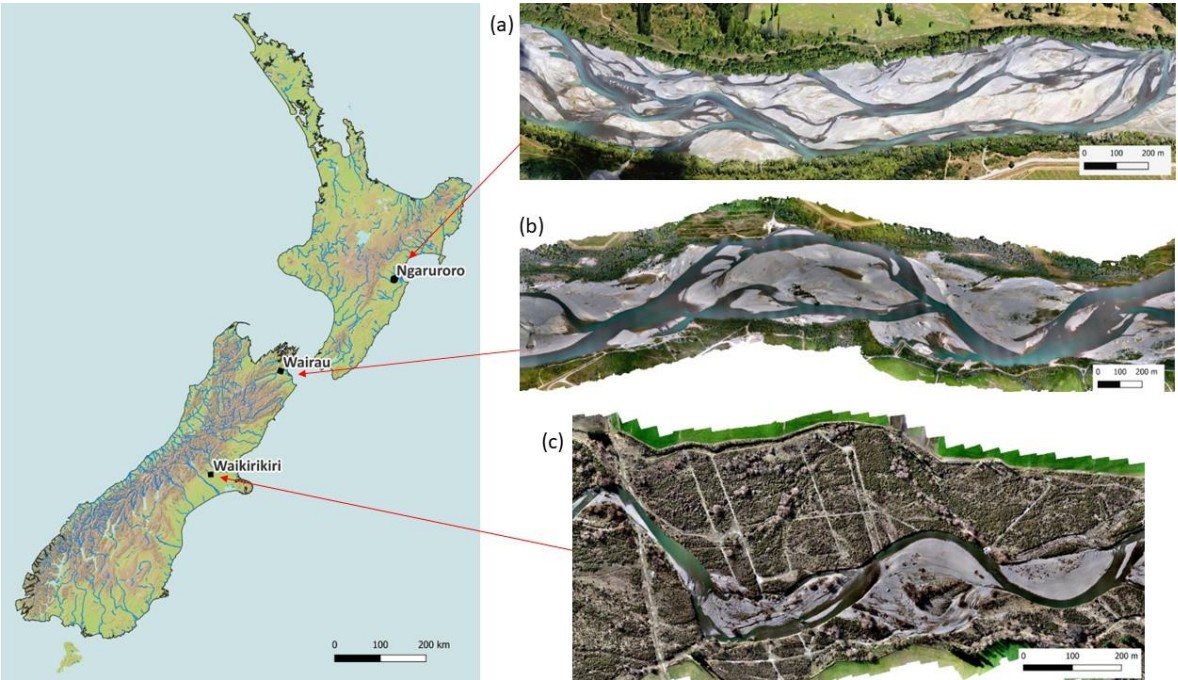

**Figure 2**. Location of the three study rivers and their aerial images

The Ngaruroro River is 164 km long, and has its headwaters in the Kaweka, Kaimanawa, and
Ruahine ranges on the main divide of the North Island. The 3 km long study reach is located
at the margin of the Heretaunga plain between Roy's Hill and Fernhill and is the main
recharge source for the Heretaunga alluvial aquifer system (Dravid and Brown, 1997). A
long-term flow monitoring site at Fernhill (70 years), situated at the lower end of our study
reach, has recorded a mean flow of 40 $m^3.s^{-1}$ and mean annual peak flow of 1546 $m^3.s^{-1}$. The
Ngaruroro study reach is in a natural depositional zone within the catchment (i.e., bedload
into the reach exceeds that leaving the reach). This reach typically has 3 braids, with the
active braidplain confined between willow plantings.

The Wairau River (Wöhling et al., 2018 and 2020) is 170 km long, and has its headwaters in
the alpine Spencer Range on the main divide of the South Island. As such, the Wairau
receives considerable source water from rain at higher elevations, producing a mean flow of
100 $m^3.s^{-1}$, and high mean annual peak flow of 1911 $m^3.s^{-1}$ (63 years of record at SH1). The
Wairau River is also "flashy", with over ten events exceeding three times the median flow
annually (FRE3, Booker 2013). The 2.2 km long study reach is in the middle section of the
Wairau Plain, between Jeffries and Giffords roads. This study reach typically has 2 braids and
the active braidplain is confined between engineered rock revetments and groynes. Bedload
flux into and out of this reach is approximately equal (bedload transfer zone).
The Waikirikiri River has its headwaters in the foothills of the Southern alps and receives
considerably less runoff than the other rivers. The 1 km long study reach is situated near
Hororata where the river leaves the foothills and crosses the western margin of the
Canterbury Plain. A long-term (58 years) flow monitoring site located upstream of the study
reach at Whitecliffs shows a mean annual flow of 4.5 $m^3.s^{-1}$, and peak annual flow of 28
$m^3.s^{-1}$. The Waikirikiri River is also the least "flashy" with an average of only one event each
year exceeding a flow of three times the median flow. As a result of its lower flow and high
transmission losses, the Waikirikiri River is subject to intermittent flow within our study
reach (Larned et al., 2008; Di Ciacca et al., 2023). The Waikirikiri study reach is naturally
incised between Pleistocene terraces, with no engineered flood controls. The active
braidplain, which typically has 1-2 braids, can freely adjust between these contemporary
braidplain margins, however, relatively dense exotic vegetation covers much of the
contemporary braidplain.

**Table 1.** Characteristics of the three study catchments. Mean values are given unless stated
otherwise. FRE3 is the annual number of events exceeding three times the median flow. Bed
material grain size is near surface but excludes the armour.

| | Ngaruroro | Wairau | Waikirikiri |
|---|---|---|---|
| Study reach length (m) | 3000 | 2200 | 1000 |
| Contemporary braidplain width in study reach (m) | 300 | 400 | 460 |
| Active braidplain width in study reach (m) | 300 | 400 | 55 |
| Braiding index in study reach | 2.95 | 1.96 | 1.56 |
| Reach slope (m.m$^{-1}$) | 0.003 | 0.003 | 0.006 |
| Catchment Area (km$^2$) | 1930 | 3320 | 250 |
| Catchment rainfall (mm.y$^{-1}$) | 1414 | 1543 | 1045 |
| Catchment potential evapotranspiration (mm.y$^{-1}$) | 722 | 693 | 750 |
| Annual low flow (m$^3$.s$^{-1}$) | 1.2 | 9.9 | 0.97 |
| Median flow (m$^3$.s$^{-1}$) | 23.1 | 61 | 2.35 |
| Mean flow (m$^3$.s$^{-1}$) | 40 | 100 | 4.5 |
| Annual peak flow (m$^3$.s$^{-1}$) | 1546 | 1911 | 28.3 |
| FRE3 | 6.1 | 10.7 | 1.1 |
| River loss at study reach (m$^3$.s$^{-1}$.km$^{-1}$) | ~0.6 | ≥0.2 | 0.25-0.65 |


All three rivers share a similar bedload lithology dominated by Jurassic greywacke. In the
study reaches, the Ngaruroro is bounded to the north by Pliocene siltstone and limestone
(Lee et al. 2011), while the Wairau is bounded to the north by schist (Begg and Johnston
2000). These two rivers lose water southwards, the river losing reaches being the primary
source of recharge for the alluvial aquifers hosted by Holocene gravels. In the Waikirikiri
study reach, the river is bounded by Pleistocene glacial outwash gravels (Forsyth et al.
2008). These gravels form the surface expression of a large, stratified aquifer system hosted
by a composite of alluvial fans which underlies the Canterbury Plains. The Ngaruroro and
Waikirikiri study reaches are both situated close to the apex of the river system's alluvial fan
(close to where the rivers emerge from the foothills).

Both the Wairau and Ngaruroro are affected by gravel extraction, which has lowered the
mean active braidplain bed elevation in the river recharge reaches by approximately one
metre since the 1980's (Gardner and Sharma 2016, Measures 2012). The varying physical
environments, flow regimes, bed adjustment trajectories, and degree of lateral confinement
result in different rates of bed reworking for each river. In particular, the Waikirikiri reworks
its bed less frequently than the Ngaruroro and Wairau.

*4.2 Field data collection*
Investigations in the study reaches involved the collection of a variety of data types (Table
2). Each method and data type has advantages and disadvantages and is scale and process
dependant (Gonzalez-Pinzon et al., 2015; Brunner et al., 2017). Stage and temperature
recorders were established in the study reaches but were difficult to maintain due to
repeated destruction by flood flows and frequent bed movement in the study reaches. Flow
rating was conducted at the top of the reach for the Waikirikiri River, and permanent rated
flow sites located downstream were used for the Ngaruroro and Wairau. A series of flow
loss gauging surveys were undertaken at multiple transects within all three study reaches
across a range of discharges to estimate transmission losses. Due to the difficulty of
measuring flow in the larger rivers (and the related large measurement uncertainty), the
most comprehensive set of flow loss surveys was conducted on the Waikirikiri (Di Ciacca et
al., 2023).

Table 2. Type and number of measurements undertaken in the three study reaches.

| Measurement type | Ngaruroro | Wairau | Waikirikiri |
|---|---|---|---|

| | | | |
|---|---|---|---|
| Differential flow gauging | 2 | 2 | 14 |
| Local river stage/temperature | | 3 | 6 |
| LiDAR and bathy surveys | 1 | 2 | 2 |
| Piezometers | 19 | 31 | 43 |
| Cored holes | 10 | 8 | 21 |
| Particle size distribution | 36 | 38 | 60 |
| Core porosity | 6 | 12 | 5 |
| Field porosity | 3 | 10 | 4 |
| Radon-222 samples | 5 | 53 | 61 |
| tTEM | Y | Y | Y |
| DualEM | Y | Y | Y |
| SkyTEM | Y | | |
| ERT surveys | | 9 | 11 |
| Ground penetrating radar (GPR) | | | 5 |
| DTS installations (vertical) | | 2 | 3 |
| DTS installations (horizontal) | | | 2 |


LiDAR data were captured in dry areas of riverbed using a LiDARUSA Snoopy LiDAR scanner
deployed on either a UAV or backpack. Bathymetry and water surface elevation were
mapped using a kayak or remote controlled jetboat equipped with a paired RTK GPS and
echosounder, and wading with an RTK GPS. Interpolation, or (where necessary) optical-
bathymetry techniques, were used generate high-resolution bathymetry maps from less-
dense echosounder survey data. The dry topography from LiDAR was stitched together with
the bathymetry data to provide a complete digital elevation model (DEM) for each reach at
a spatial resolution of 1 m or less, and a vertical accuracy of ±0.1 m in dry areas and ±0.2 m
in wet areas.
Piezometers were installed at different depths to provide a time series of water levels and
temperature, and to enable sampling for radon-222 analysis. The piezometers were
installed with 50mm diameter PVC. Screens were a 1-2m length of slotted casing with a
geotextile sock and sand pack around the screen, and cement grout around the overlying
casing. A sump was used to collect downward percolating water in situations where pore
water pressure was below saturation. Drilling methods involved a mix of rotary and sonic
drilling, the former being used to install the piezometer network. Sonic drilling with a
Geoprobe 8140LC (76.2mm diameter core) was conducted to collect cores for detailed
logging and sediment analysis (grainsize and porosity). The sonic drilling method is not ideal
as it can be difficult to get good core recovery, particularly in coarser or loose near-surface
sediments. As a result, the core record is incomplete, however, there is currently no better
method available to extract several meters of core from coarse riverbed deposits. Because
core recovery was variable, data from individual drillholes were fragmented, and sediment
analysis was only conducted on the most intact cores (0.1-1.2 m in length), with samples
taken for grainsize and porosity analysis. Near-surface bed material grain size distribution
and porosity were measured using manual excavation sieving combined with a detailed
photogrammetry method to calculate excavation volume (Montgomery et al. subm.).
Particle size distribution was characterised by fitting Gompertz and Weibull curves (Bayat et
al. 2015) to the sieved sediment data, and summary statistics determined via the method of
Folk and Ward (1957).
The base of the BPA was identified in two ways. The first method was by mapping surface
morphology, since the deepest areas of pools represent the scouring depth that has
occurred during flooding. Surface and bed morphology was captured using remote sensing
(photogrammetry, LiDAR, and bathymetry). The second method was through observations
in drill core with good core recovery, where sediments beneath the BPA are characterised
by more cohesion and a finer matrix. To enable data to be compared, contacts between the
braidplain gravels and underlying sediments (depositional unconformity) have been
expressed as depth below a detrended surface representing cross-section mean
contemporary braidplain surface elevation. This datum was chosen because of the spatially
variable and temporally dynamic river bed levels.
Hydrogeophysical methods were also used to image the subsurface, including passive (DTS)
and active (ADTS) distributed temperature sensing (Banks et al., 2022), ground penetrating
radar (GPR), electrical resistivity tomography (ERT), transient electromagnetic (tTEM) and
electromagnetic induction (DualEM421). The tracks prepared for tTEM and DualEM surveys
are evident in the aerial photos shown in Figure 3. SkyTEM data were also available for the
Ngaruroro area (Rawlinson et al. 2021). Of the hydrogeophysical methods employed,
DTS/ADTS, and ERT were the most successful methods for delineating sediment structure
and saturation associated with the river. The resistivity of New Zealand braided river water
(fluid specific conductance ~5 mS.m$^{-1}$) and associated gravel deposits is very high (400-
10,000 Ωm). For this reason, we think there was insufficient resistivity contrast for
electromagnetic and ERT methods to reveal distinct subsurface features in most of our
surveys. SkyTEM data did provide good definition of the basement contact beneath the
Ngaruroro River but did not reveal any clear structural features in the near surface (<10 m).
GPR surveys that were trialled at the Waikirikiri site clearly revealed the shallow water table
but did not reveal any clear structure beneath the water table due to reflection of the
signal.
Samples for radon-222 analysis were collected in 250 ml bottles in the Waikirikiri (Songola
2022) and Wairau reaches from riffles, at different depths in pools, and from purged
piezometers. Samples were analysed 21-100 hours after sampling for Waikirikiri, and 20-24
hours for the Wairau. Laboratory analysis for radon activity was conducted with a RAD7
(Durridge 2020a), RAD H2O (Durridge 2020b), and active DRYSTIK (Durridge 2021) in a
closed loop system, with the results adjusted for decay since the time of sampling (WAT250
method). The radon activity and uncertainty values reported here follow the approach of
Durejka et al. (2019), with the mean and standard deviation calculated from five counting
cycles, with duplicate samples pooled (ten cycles total). Additional radon measurements
were made in the field using the RAD AQUA method (Durridge 2020c) to verify the WAT250
method results, and these returned similar values. For this study we have reported the
WAT250 data, which has a larger uncertainty associated with the measurements but
enables more samples to be collected in a short time frame from remote field sites. To
reduce the uncertainty of the WAT250 results, we increased the aeration time to 10
minutes, and the analysis duration recommended in the Durridge manual to 5 cycles of 10
minutes.

**5.0 Field observations**
A hydraulically disconnected river-regional aquifer system has been identified from drilling
and monitoring data in the Waikirikiri study reach (Banks et al. 2022, Di Ciacca et al. 2023),
and at the upper part of the Ngaruroro study reach. A hydraulically connected river-regional
aquifer system is observed in the Wairau study reach, and in the lower part of the
Ngaruroro study reach. Evidence for the proposed conceptualisation is provided below
based on data type. Data referred to in the text and figures is shown spatially in Figure 3.
Points where the base of the BPA has been observed in cores, or inferred via deep pools are
identified.

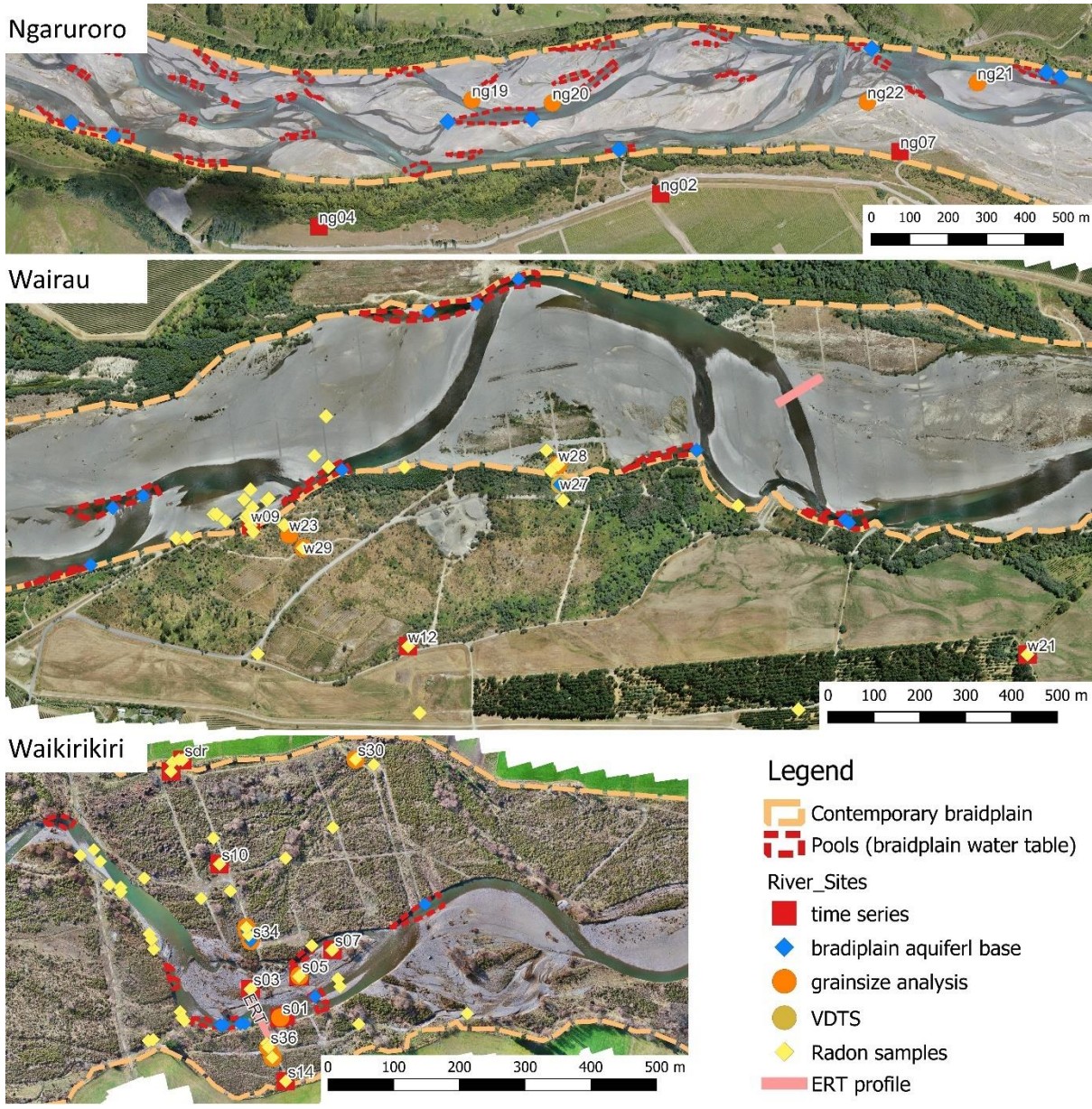


**Figure 3.** Data sources referred to in the text.


*5.1 Geology and geomorphology*

The BPA lateral extent in our three sites is identified geomorphologically as being at the

contemporary braidplain margins (orange dashed lines in Figure 3). The contemporary

braidplain margins in our study reaches are either artificially confined by rock armouring or

willow planting for flood protection (Wairau and Ngaruroro), or by terrace boundaries

(Waikirikiri). These relatively static lateral controls result in the abutment of relatively loose

(recently active) braidplain gravels against older, more poorly sorted sediments. In the
Wairau and Ngaruroro rivers the artificial lateral controls are sufficiently narrow that the
entire contemporary braidplain is regularly reworked (i.e., the active braidplain margins and
contemporary braidplain margins are essentially the same). In the Waikirikiri, the active
braidplain is narrower than the contemporary braidplain and the active braidplain is able to
adjust laterally, reworking the contemporary braidplain gravels.
In all three braided rivers, deep pools form at the toe of riffles. In the Wairau and
Ngaruroro, these are amplified at the braidplain margins where river flow is reflected by
rock armouring or willow plantings that limit lateral erosion and enhance bed scour. These
pools are zones where the BPA discharges to the river. Parafluvial seeps are commonly
observed in these locations, which have higher radon-222 activities and summer
temperatures that are cooler than adjacent river braids. The surface expression of the BPA
can be seen in abandoned channels where the braidplain surface topography drops below
the braidplain water table and exposes groundwater (static pools or flowing springs). These
areas are demarked by dashed red lines in Figure 3.
The depositional unconformity at the BPA base in recovered cores was observed at 1.7-2.0
m depth beneath the mean bed level of the contemporary braidplain in the Ngaruroro study
reach, 4.3-5.0 m depth in the Wairau, and 2.1-3.7 m depth in the Waikirikiri. The greater
depth in the Wairau River is unsurprising, as this river has the highest flood flows and is
tightly confined by engineering works, resulting in greater depths of bed reworking.
At the Waikirikiri site, the contact of the depositional unconformity was seen as a change
from grey-brown postglacial sandy gravels to yellow-brown clay-rich glacial outwash gravels
(Figure 4). During drilling and completion of the groundwater monitoring wells, a drop in the
water level with depth below the glacial contact was observed. Six cores were drilled just
beneath the unconformity at various locations within the contemporary braidplain margins
to investigate the saturation status beneath the braidplain gravels. These holes were drilled
through the low permeability contact layer until an apparent increase in permeability was
encountered, at which point the water level in the drillhole dropped significantly.
Piezometers screened beneath the base of the BPA and above the regional water table
showed low and unvarying water levels, indicating unsaturated or variably saturated
conditions (Figure 7). Our explanation for the presence of this variably saturated zone is that
the glacial outwash gravels are stratified, and vertical infiltration to the regional aquifer is
limited by the lower permeability horizons of silt and clay. The presence of higher
permeability horizons within the stratified postglacial sequence allows water to move
laterally away from the recharge zone at a rate that exceeds vertical infiltration, enabling
unsaturated or variably saturated conditions to form.

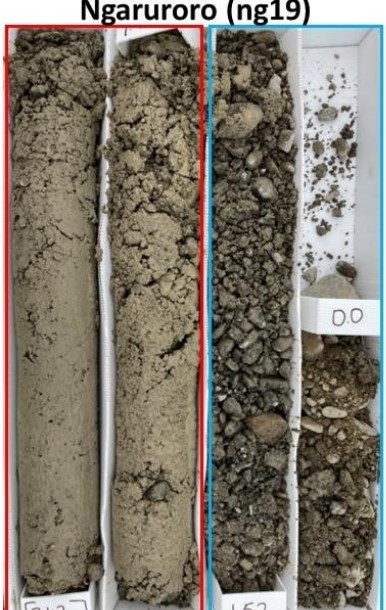 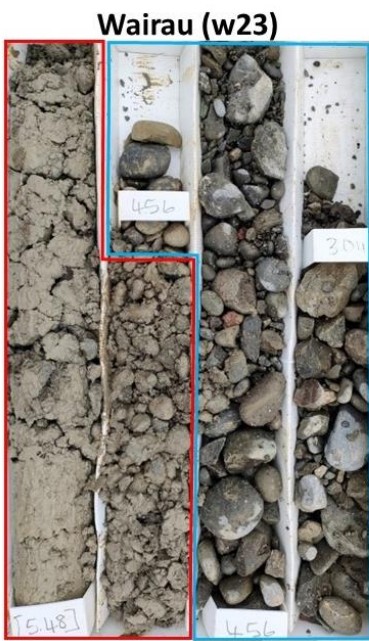 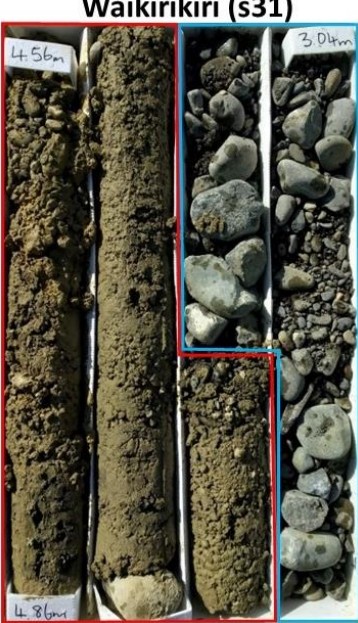

**Figure 4.** Representative core samples across the unconformity at the three study sites
(blue=braidplain gravels, red=underlying very poorly sorted consolidated gravels)

At the Wairau River study site, the base of the BPA is visible as a change from grey-brown

sandy clast supported gravel to more cohesive and poorly sorted gravel with increasing

proportions of silt and clay. Just beneath this unconformity is a more prominent contact

with yellow brown silty clay-bound gravel associated with old outwash fan deposits along

the Richmond Range (not shown in Figure 4, but evident in Figure 10). Note that some of

the Wairau core holes were positioned on the berms (outside of what we are referring to as

the contemporary braidplain), where the active braidplain was located prior to river re-

alignment in the 1960s. The river can no longer mobilise sediments on these berms due to

rock armouring of the banks leaving a recent remnant braidplain gravel deposit beneath the

southern berm. This deposit is both spatially and vertically separate from the contemporary

braidplain (the mean bed level of the contemporary braidplain is approximately 2m below

the berms).

In the Ngaruroro study site, the unconformity at base of the BPA is more gradational,

manifesting as an increase in silt and clay content between 1.7 and 2.0 m depth visible

within bore logs (Figure 4) and particle size distribution (Figure 5). As with the Waikirikiri

River, a drop in the water table was observed while drilling beneath this lower permeability

unconformity. The less-distinct base to the BPA in the Ngaruroro study reach is likely

because the reach is depositional, so the underlying gravels were deposited relatively

recently compared to the other study reaches.

Particle size distribution summary statistics of core samples collected by sonic drilling and

bed excavation are shown in Figure 5, with classifications according to the Folk and Ward

(1957) scale. A relationship between sorting and grainsize is apparent at each site, with

poorer sorting corresponding to an increase in the finer fraction. Shallow braidplain gravels

are overall poorly sorted (1-2, data mean: Ngaruroro 1.98, Wairau 1.90, Waikirikiri 1.96),
whereas the underlying gravels tend to be very poorly sorted (2-4, data mean: Ngaruroro
2.88, Wairau 2.28, Waikirikiri 2.60). The grainsize of the braidplain gravels is generally
coarser (mean: Ngaruroro -4.76, Wairau -4.23, Waikirikiri -3.55) than the underlying very
poorly sorted sediments (mean: Ngaruroro -1.64, Wairau -3.54, Waikirikiri -2.69).

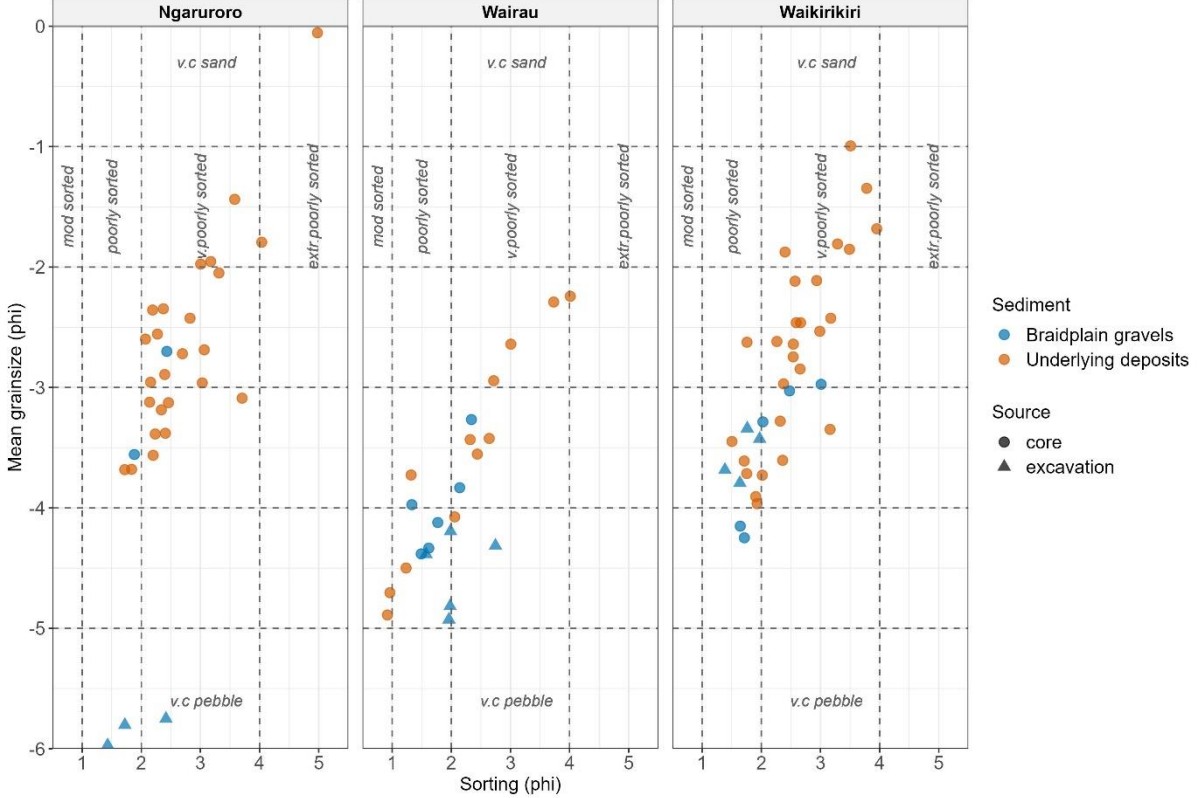


**Figure 5.** Grainsize analyses and sorting indices from samples beneath the three study
reaches. Dashed lines are breaks for grainsize and sorting indices (after Folk and Ward

561    1957).

Core recovery at all sites was typically poor in the contemporary braidplain gravels, which
mostly consist of loose gravel and cobbles, with a high proportion of sand but only a trace of
silt and clay. Because of this, some of the braidplain gravel samples have been sourced from
samples manually excavated from the braidplain subsurface (Figure 5). These excavated
samples exclude the surface armour common in gravel bed rivers. For the Wairau data set
there is no clear separation in the particle size distribution between braidplain and
underlying gravels. Wairau sediments are coarser than the other two sites, because of the
high-energy of this river system. The coarseness of the sediments greatly hampers core
recovery, so it is possible that the lack of separation is an artefact of the loss of finer
fractions in some samples during drilling.

*5.2 LiDAR and bathymetry*
LiDAR and bathymetry surveys were conducted in each study area to understand the
spatially varying relationship between the river surface, bed levels, and water levels in the
braidplain and regional aquifers. Repeat surveys were conducted following significant flood
events to capture changes in bed levels. An example of our LiDAR and bed elevation data for
the Wairau River is shown in Figure 6. These data were captured on 19 Feb 2020 at
relatively low flow conditions, measured at 13.4 $m^3.s^{-1}$ to 11.5 $m^3.s^{-1}$ (±3 %) at the upstream
(left) and downstream (right) margins of Figure 6 respectively. The river water surface and
bed elevation data within the wetted channel are shown on Figure 6 in relation to a
modelled surface of hydraulic head across the river system, represented by piezometric
contours which are shown in Figure 6a. This surface was fitted (sum squared error of 5 x 10⁻
⁶) to 25 water level observations (yellow points in Fig 6a) located within and outside of the
contemporary braidplain by universal kriging with an exponential variogram of anisotropy of
0.9 at 090°, partial sill 0.31 m, and range 670 m. With such a large variogram range, the
surface should be considered as indicative of an averaged hydraulic head across the regional
and braidplain aquifers. The kriged surface does reveal an inflection of the piezometric
contours across the contemporary braidplain margins, indicating that flow within the BPA is
largely controlled by river exchange and preferential flow within the BPA, with flow being
approximately sub-parallel to the contemporary braidplain longitudinal orientation.
Fig. 6a reveals locations in the river system where the river water surface is higher than the
braidplain water table (red and orange zones), indicating that the river is losing flow to the
BPA in these areas. Areas of the river which are coloured blue in Fig. 6a represent the
surface expression of the braidplain water table in pools. These are locations where the
river can potentially gain flow. The black areas denoted as "riffles" are identified from a
slope raster derived from the digital elevation model (DEM). Locations where maximum
potential river water loss occurs can be identified in most cases as being situated at the
upstream margins of high elevation riffles.
The bathymetry DEM (Fig. 6b) reveals the presence of scouring along the contemporary
braidplain margins, which in the case of the Wairau River is promoted by excessive river
narrowing and rock training banks. The corollary of this scouring is the relative mounding of
gravel in the middle of the contemporary braidplain. The difference between the river bed
level and hydraulic head reveals locations where the river bed is above the braidplain
aquifer, and the river braid has the potential to be losing-disconnected at these locations. In
most cases these areas also correspond to the upstream margins of high elevation riffles.

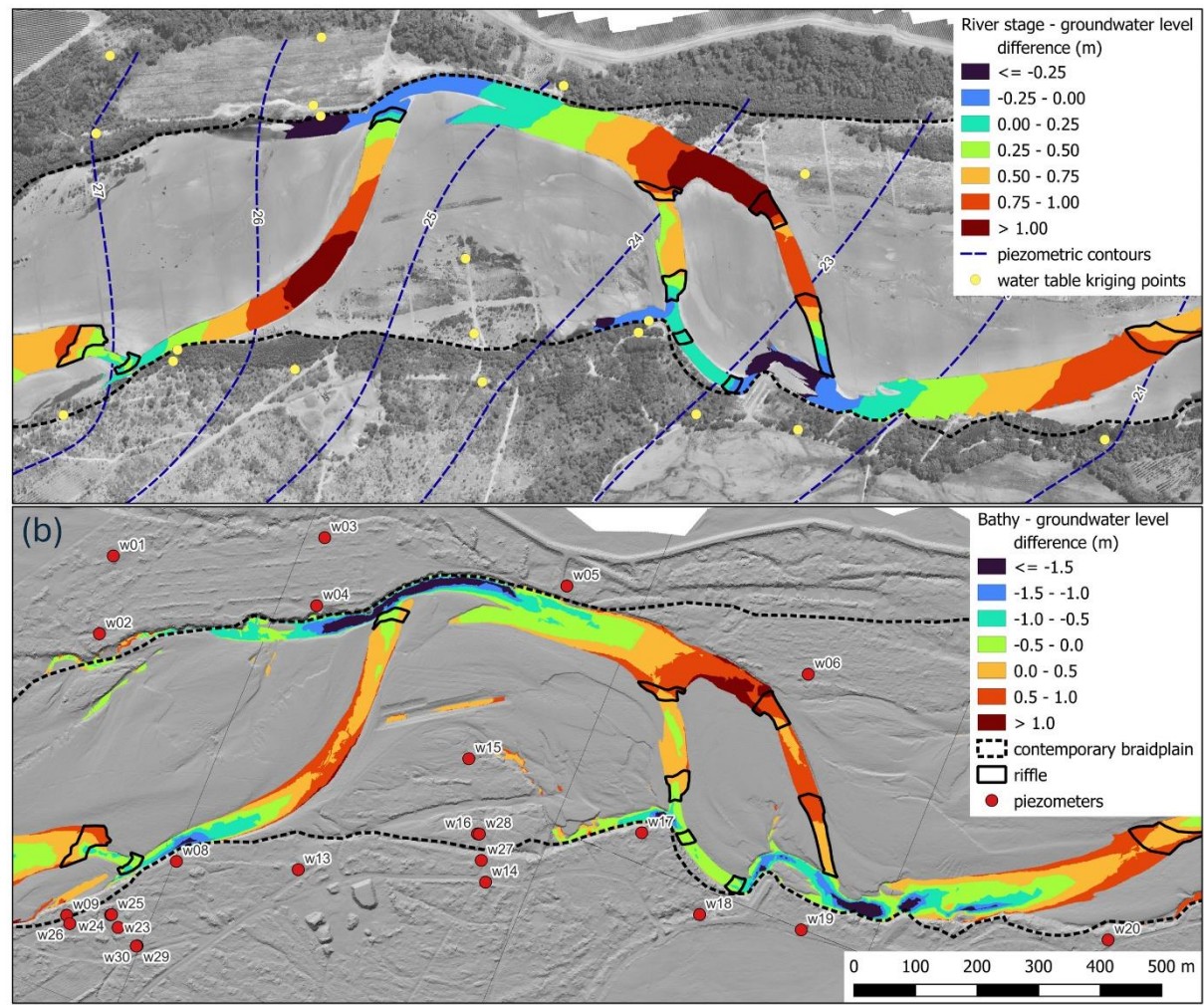


**Figure 6.** Images of the wetted Wairau River channel showing differences between (a) river

water surface and a kriged hydraulic head (overlain on aerial imagery), and (b) bathymetry

and kriged hydraulic head (overlain on the DEM). River flow is from left to right.

*5.3 Water levels and temperature monitoring*

Hydraulic heads within the braidplain aquifer are dynamic and fluctuate in response to

changes in river stage. Figure 7 shows heads for the Waikirikiri field site, where the base of

the BPA is at approximately 207m elevation, and the regional water table is >13m below the

unconformity. The variably saturated zone (pore water pressure below saturation) is at least

10m thick. For a river system that is hydraulically connected to the regional aquifer, the

pressure response outside of the BPA is also dynamic and shows a similar response to the

BPA (Figure 7). For a river system with a hydraulic disconnection, the variably saturated
zone attenuates the pressure fluctuations in the regional aquifer (Figure 7). The relative
water level depth and hydraulic response in the regional aquifer can therefore provide a
useful test for hydraulic connectivity between the two aquifer systems.

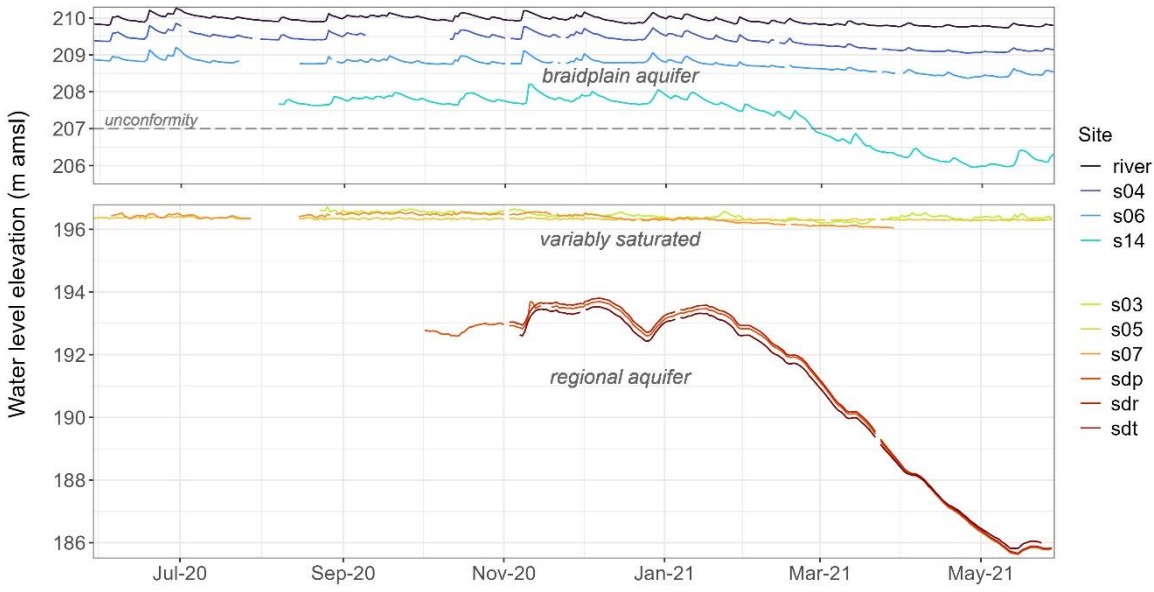


**Figure 7.** Water level time series data for the Waikirikiri River and braidplain and regional
aquifers. Note the vertical offset in the two graphs is due to the thick variably saturated
zone beneath the braidplain and regional aquifer. Site locations are shown in Figure 3.

Data from the Ngaruroro reach show characteristics of a regional aquifer that is both
hydraulically connected to the BPA and either disconnected or transitional (Figure 8). These
data have been normalised by the median water level to highlight relative changes in
response. The BPA site (ng07) responds similarly to the river stage but has a slower
recession rate due to storage in the gravels. The two sites that are screened in the regional
aquifer (ng02, ng04) are both located 150 m adjacent to and downgradient of the
contemporary braidplain (see Figure 3). The upstream site, ng04, has the most rapid
recession rate and its peak response is slightly delayed compared to ng02 which responds
similarly to the river channel stage. This suggests that this upstream section of the study
reach may be hydraulically disconnected or transitional from the regional aquifer (assuming
similar hydraulic properties throughout the regional aquifer). The water levels in ng04 are
up to 2.4 m below the BPA base (mean 1.8 m) assuming a BPA thickness of 1.6 m below
mean river bed level at this site. During flood flow events, the regional aquifer water level
ng04 does rise above the river bed elevation, indicating saturated conditions do temporarily
occur during flooding.

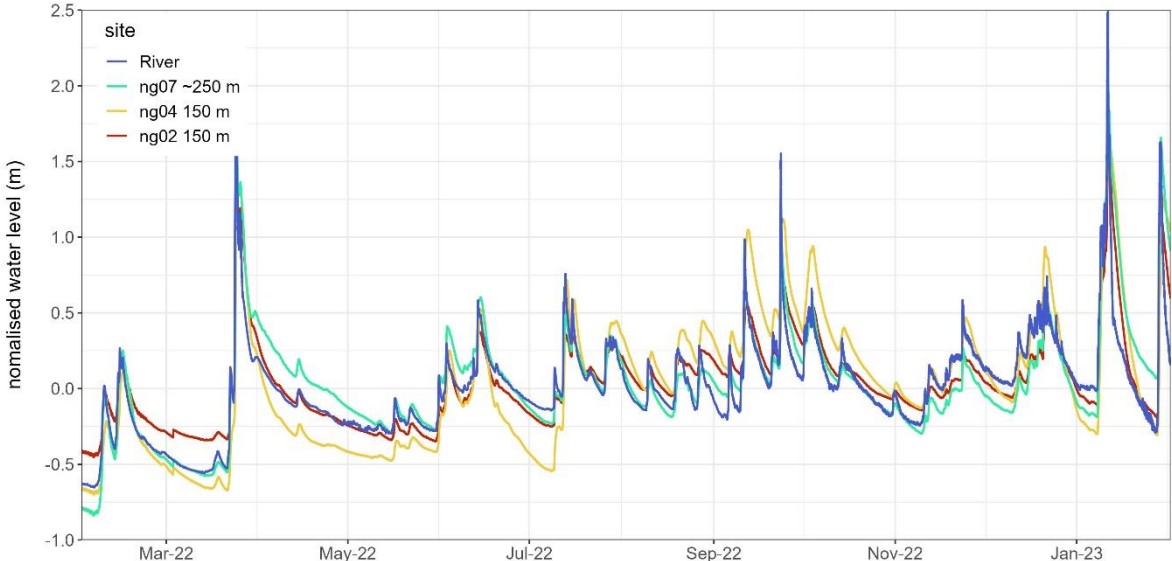


**Figure 8.** Normalised water level time series data for the Ngaruroro River, braidplain (ng07)
and regional aquifers (ng02, ng04). Site locations are shown in Figure 3. Distances in the
legend indicate the down gradient distance from the nearest active river channel.

Figure 9 shows daily average temperatures from representative hydrological settings in each
study reach, and the lateral distance downgradient of the nearest active river channel. For
sites located within the BPA, temperature responses are similar to the river channel, and
driven by individual flow events (ng07, w09, s06). However, event-driven responses can be
attenuated, and the season response delayed, at large distances from a channel (s25). Note
also that s06 becomes more responsive after a flood event on 30 May 2021 which has
changed connectivity between the river and groundwater in this piezometer.
For sites within the regional aquifer (ng02, ng04, w12, w21, sdp), the event-driven response
is difficult to detect, and the seasonal response depends on the hydraulic relationship
between the braidplain and regional aquifers. Sites ng02 and ng04 are equidistant from the
Ngaruroro BPA, however ng04 is in a position where the base of the braidplain gravels and
regional water table are approximately 1.8m apart, potentially with an intervening variably
saturated zone, whereas at ng02 the braidplain and regional aquifers are hydraulically
connected. This difference in hydrologic condition may explain the delayed seasonal
response in ng04. In the Waikirikiri system, the variably saturated zone is considerably
thicker (about 12m), which almost completely attenuates the seasonal temperature
response in the regional aquifer.

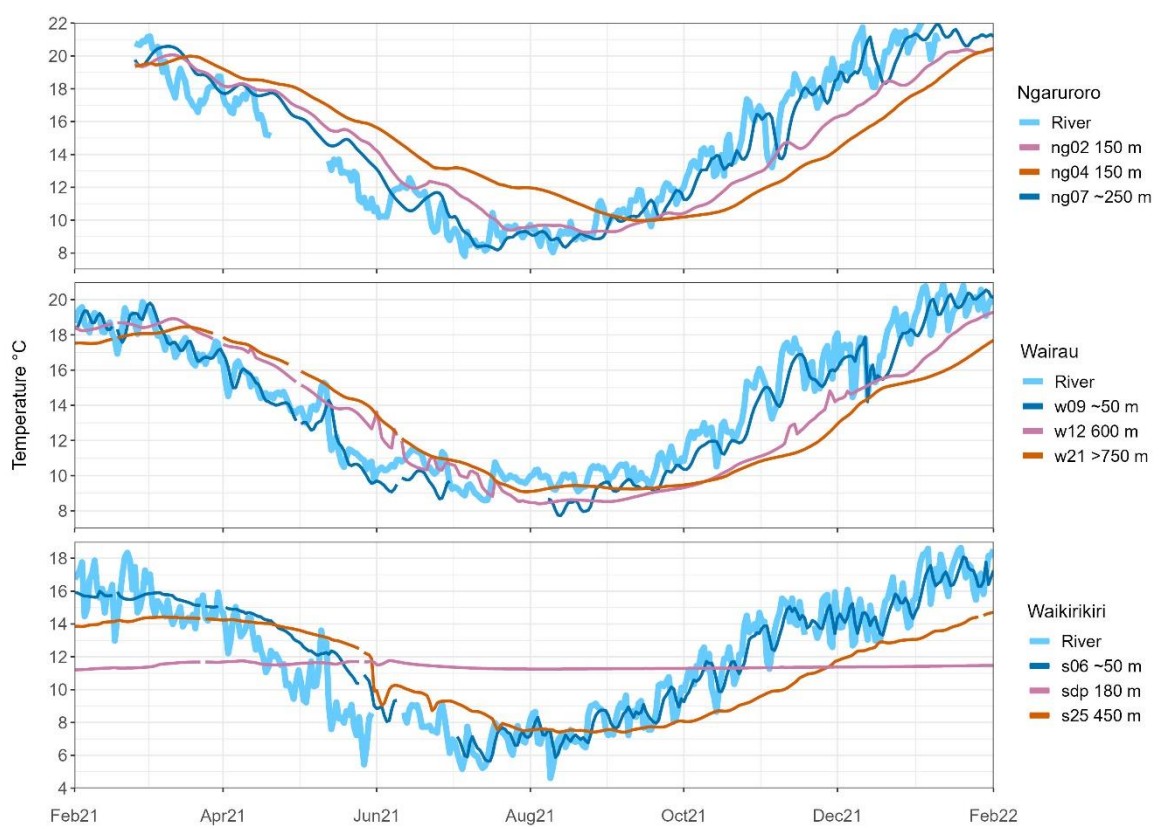


**Figure 9.** Representative temperature time series for the three study sites. Piezometer
locations are shown in Figure 3, sites ng07, w09, s06 and s25 are screened in the BPA.

In the Waikirikiri and Ngaruroro river systems, the temperature signals propagate efficiently
from river channels to the BPA compared to the regional aquifer. This highlights the
distinction between these two aquifers, with the BPA acting as an intermediary in
groundwater – surface water exchanges.

*5.4 Radon-222 sampling*
A summary of the radon-222 results and measurement uncertainties for surface water and
groundwater sources in the Wairau and Waikirikiri reaches is shown in Table 3. In the
Wairau system, samples from riverbed piezometers and riverbed seepages have similar
radon activities with ranges 1724-2849 BQ.m$^{-3}$ and 368-2585 BQ.m$^{-3}$ respectively.
Accordingly, samples from seepages and riverbed piezometers are both considered to
represent the braidplain aquifer.
The radon data show distinct groupings, with radon-222 activities increasing from river
channel to BPA to regional aquifer. At both sites, radon activities in river run samples were
significantly lower than those in BPA samples. In the Wairau study reach, there is a notable
overlap in radon activities between the braidplain and regional aquifers, indicating a likely
hydraulic connection between these two systems. Conversely, in the Waikirikiri study reach,
there is a downward increase in radon activities from the BPA to the variably saturated zone
and further into the regional aquifer, with no overlapping values. This suggests a hydraulic
disconnection between the BPA and the regional aquifer in the Waikirikiri reach.

**Table 3.** Measured radon-222 activities and one standard deviation uncertainties (BQ.m$^{-3}$)
for the Wairau and Waikirikiri study reaches.

| River | Wairau | | | Waikirikiri | | | |
|---|---|---|---|---|---|---|---|
| Sample source | River run | Braidplain aquifer | Regional aquifer | River run | Braidplain aquifer | Variably saturated glacial outwash sediments | Regional aquifer |
| Samples | 14 | 11 | 21 | 10 | 38 | 7 | 6 |
| Rn min | 260 ± 85 | 368 ± 150 | 739 ± 272 | 384 ± 185 | 1791 ± 481 | 7017± 3607 | 16111 ± 3305 |
| Rn max | 604 ± 224 | 2849 ± 826 | 5700 ± 629 | 809 ± 443 | 9545 ± 1378 | 14654 ± 4338 | 22655 ± 2221 |
| Rn mean | 395 ± 180 | 1307 ± 361 | 3263 ± 666 | 568 ± 356 | 4569 ± 1002 | 11814 ± 2589 | 19184 ± 2763 |


The determination of residence times between the river and each aquifer depends on
knowing the initial channel condition, representative secular equilibrium for the host gravel
deposit, as well as a well-defined flow path length. Our estimate of the initial river channel
condition is 260 BQ.m$^{-3}$ for Wairau and 380 BQ.m$^{-3}$ for Waikirikiri, reflecting the lowest
measured river radon-222 activities. A secular equilibrium estimate of 5000 BQ.m$^{-3}$ was
derived for Wairau aquifer samples by plotting measured groundwater radon activity
against distance of the piezometer from the river and fitting the ingrowth equation to the
data to determine the 21 day equilibrium value. This exercise indicated that the Wairau BPA
activities are too low for samples to reach equilibrium. In the absence of sediment specific
data, the Wairau aquifer secular equilibrium was chosen to represent the Wairau BPA
equilibrium. The secular equilibrium for the Waikirikiri BPA is estimated at approximately
8500 BQ.m$^{-3}$ based on the lowest activity observed in porewater samples from piezometer
sumps in the variably saturated glacial outwash gravels beneath the braidplain aquifer (7017
BQ.m$^{-3}$), and the highest activity observed in the BPA (9545 BQ.m$^{-3}$). Based on the secular
equilibrium values chosen, residence times for our study reach samples are estimated to be
in the range of 0.1 to 4.4 days for the Wairau BPA in the study reach and 1 to >12 days for
the Waikirikiri BPA. Due to the large uncertainties associated with the WAT250 method,
these estimates should be considered for comparative purposes only.

*5.5 Geophysics*
ERT surveys of the contemporary braidplain in the Wairau and Waikirikiri reaches yielded
varying degrees of success. The surveys that returned the most consistent subsurface
resistivity response were conducted in the river channel itself, due to the presence of water
in the near surface, which improved the connection between electrodes and the underlying
resistive substrate. Figure 10 shows dipole-dipole array resistivity profiles across a braid of
the Wairau (~ 30 m wide), and two braids of the Waikirikiri (~40 m wide). These profiles
reveal the contact between the braidplain aquifer and underlying sediments (the
unconformities shown in Fig. 4) at elevations consistent with drillhole coring. For
comparison, the elevation of the same unconformity, derived by kriging intercepts of the
BPA base within the contemporary braidplain (Figure 3.), is shown in red in Figure 10. The

two profiles are shown at the same spatial and resistivity scale, and an interpretation has

been made based on drilling information. Both profiles show a contrast between resistive

(~1000 ohm-m) loose sandy gravels that host the BPA, and the underlying lower resistivity

(<800 ohm-m) associated with older, very poorly sorted sediments. The Wairau profile

reveals the unconformity at the base of the active braidplain gravels as a less resistive layer

at ~19.5 m elevation, and a saturated BPA thickness of ~3.5 m. On the Waikirikiri profile, the

unconformity at the base of the braidplain gravels lies at ~207.5 m elevation, and the

saturated thickness of the BPA is only 2-2.5 m. The underlying very poorly sorted glacial

outwash gravels typically show relatively low resistivities (≤ 600 ohm-m) due to the relative

abundance of clay-sized sediment, even when not fully saturated.

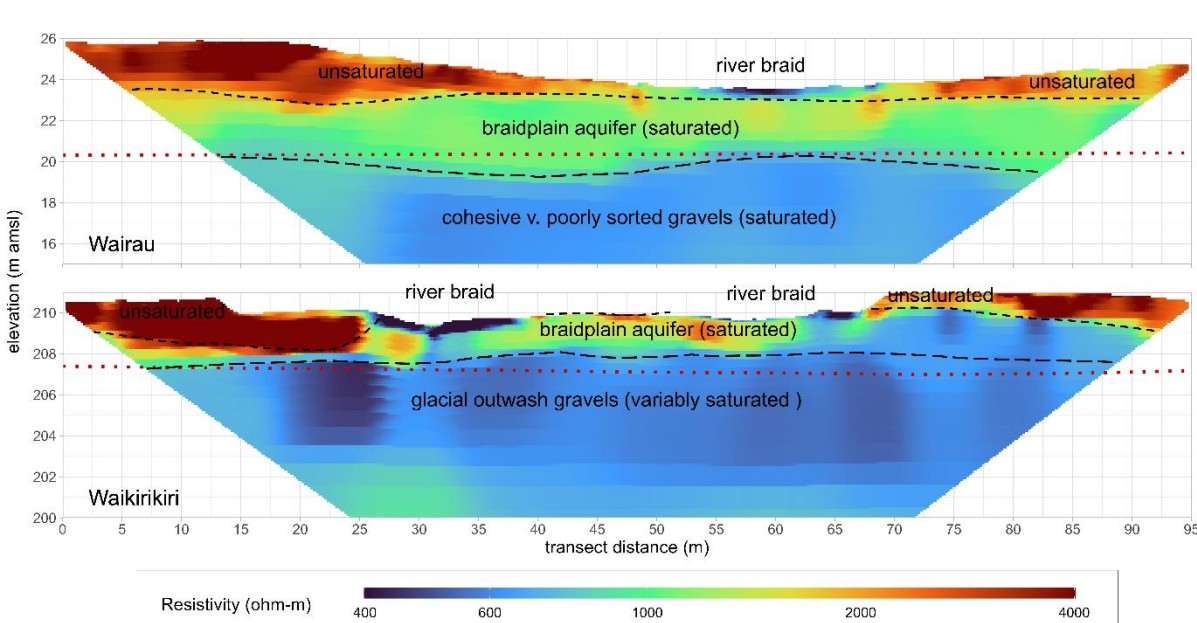

**Figure 10.** Subsurface resistivity collected by Electrical Resistivity Tomography (ERT) across

Wairau and Waikirikiri river braids, looking downstream. Surfaces interpreted from the

resistivity data are shown in black. An interpolated surface for the base of the BPA derived

from drilling and bed data is shown in red.

*5.6 Passive and active distributed temperature sensing*

The hydrogeologic structure in the Wairau study reach has been assessed by monthly DTS

and ADTS surveys (Figure 11) conducted on a vertically installed fibre optic cable located 20

m from the active braidplain margin (w27 in Figure 3). While w27 lies just outside of the

existing engineered contemporary braidplain, this site does contain remnant braidplain

sediments deposited prior to stabilisation of the river margins in the 1960s. The timing of

the DTS surveys with respect to river temperature is shown in Figure 11c. The survey

temperature profiles show consistent inflections with depth due to the attenuation of the

river recharge temperature response through saturated sediments of contrasting hydraulic

conductivity (Figure 11). These inflections correspond to the depths where a change in

sediment characteristics can be seen in the bore log, manifest as progressively increased

sediment cohesion, poorer sorting, and increasing silt and clay content with depth. In the

DTS profile, the older silty clay-bound and indurated sediments have a large influence on

subsurface flow, i.e., a reduction in flow through this material.

756

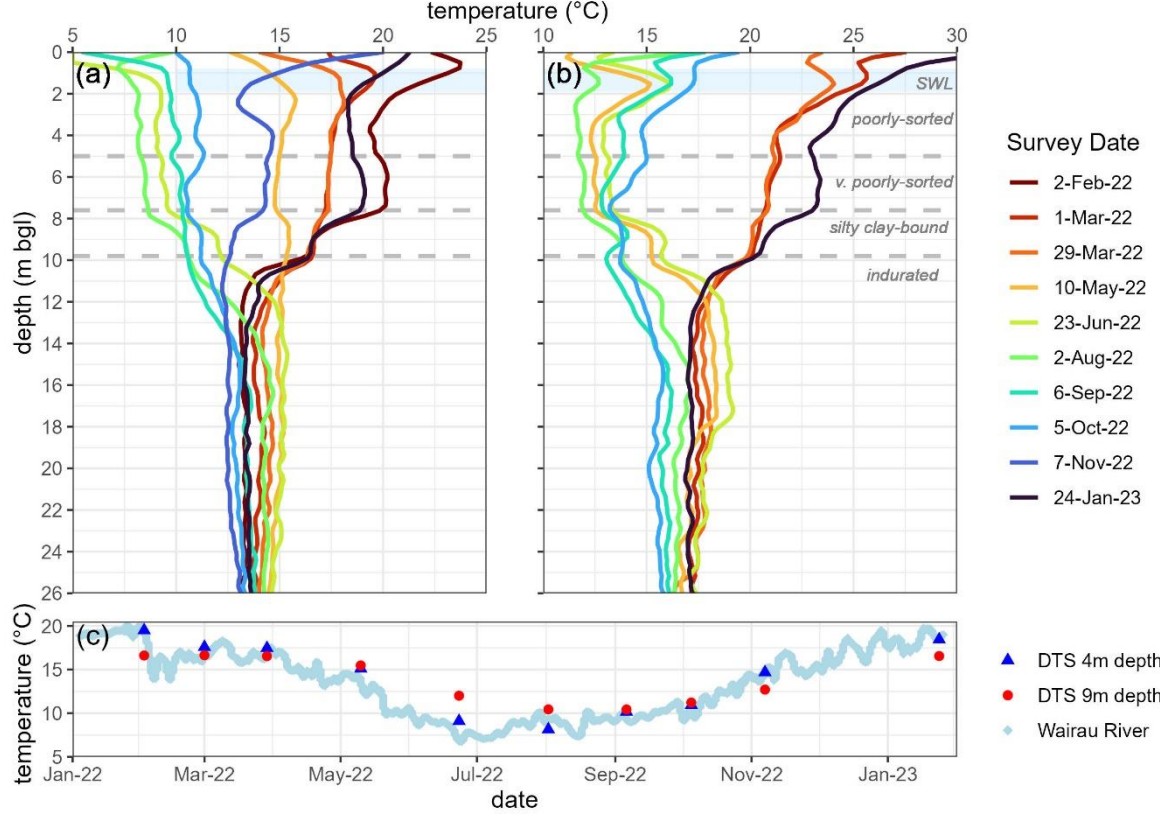

757

**Figure 11.** Vertical (a) distributed temperature sensing (DTS) and (b) active-distributed

temperature sensing (A-DTS) surveys conducted on the south bank of the Wairau River on

w27. The river temperature over the period of the surveys is shown in (c) along with

temperatures measured by DTS at 4 and 9 m depth. SWL=static water level measurements

over the survey period

763

**6.0 Characteristics of the identified braidplain aquifers**

A summary of the approximate dimensions and potential maximum groundwater storage

volumes of the three braidplain aquifers investigated is shown in Table 4. In all three

reaches, the BPA is laterally extensive, but very thin.

The Wairau River has the greatest BPA storage potential, largely due to its observed
saturated thickness of up to 4.1 m. However, the Wairau is also a highly channelised river,
and has a large bathymetric range, giving its BPA a large potential variability in saturated
thickness in response to river channel stage fluctuations. Of significance for water
management in the Wairau Plain is that the very poorly sorted gravels are thin beneath the
Wairau contemporary braidplain and underlain by buried fans of silty clay-bound and
indurated gravels which act as a vertical flow barrier (see Figure 10). Gravel extraction from
the river system has dropped the mean bed level by approximately 1 m within our study
reach since the early 1990s (Gardner and Sharma, 2016). This has allowed the river to
rework sediment to greater depths, which has thinned and reduced the effective
transmissivity of the gravel sequence overlying the buried fan deposits in addition to
reducing the hydraulic gradient between the river system and regional aquifer.

**Table 4.** Dimensions and maximum storage volumes of braidplain aquifers observed in the
three study areas.

| River | Approximate width (m) | Contact depth (m) min-max (mean) | Porosity min-max (mean) | Storage volume (m$^3$.m$^{-1}$) |
|-------|----------------------|----------------------------------|-------------------------|-------------------------------|
| Ngaruroro | 300 | 1.2 - 2.4 (1.6) n=4 | 0.12-0.35 (0.21) n=9 | 80-150 |
| Wairau | 450 | 2.4– 5.7 (4.1) n=6 | 0.09-0.46 (0.22) n=21 | 210-500 |
| Waikirikiri | 400 | 2.0 - 3.7 (2.8) n=16 | 0.09-0.22 (0.14) n=9 | 130-240 |


**7.0 Discussion**
The proposed conceptualisation enables a complex braided river system of high complexity
to be represented by a few key hydrogeological elements. These are represented in Figure 1
as a contemporary braidplain, the margins of which mark the lateral extent of the braidplain
aquifer (1), an underlying unconformity with more consolidated and more poorly sorted
sediments (2), and observations of the hydraulic relationship between the braidplain and
regional aquifers (a or b). For braided rivers, this approach integrates existing concepts of an
alluvial aquifer, and hydrological hyporheic and parafluvial zones into a single
conceptualisation of the contemporary braidplain subsurface. By identifying the base and
margins of the BPA, and the process which forms it (reworking of bed material), the vertical
and lateral extents to which hyporheic and parafluvial exchange occur can be identified by a
change in sediment characteristics.

Point-scale features of braided rivers are described well by the existing hydrological
framework proposed by Fox and Durnford (2003) and modified by Brunner et al. (2009a and
b). Under this framework, individual channels can be considered as hydrologically connected
(gaining or losing), disconnected (losing), or transitional (losing). However, this framework
starts to break down for braided rivers at local to sub-catchment scales, as all four of these
hydrological states can occur along a single cross-section of the braidplain regardless of
whether the river has a net gain or loss. However, at the reach or sub-catchment scale, a
braided river can be considered a *river system*, which can be described by any one of those
hydrological states in relation to the regional groundwater system. For example, a river
braidplain reach can be hydrologically disconnected and be losing water to groundwater
overall, even though individual channels are hydrologically connected and locally gaining
flow from BPA groundwater. The conceptualisation posed by Fox and Durnford (2003) can
therefore be applied to different scales within a braided river system, but its application,
and therefore interpretation of field measurements, requires knowledge of subsurface
structure and saturation.

The difference in sediment characteristics above and below the unconformity at the base of
the BPA indicates that the process of BPA formation is controlled by the mobilisation of bed
material during flood flows, which loosen and sort the braidplain gravels, and winnow the
finer fraction. This process of gravel mobility associated with flood events is supported by
bathymetric observations of the depth of river channel scouring in deeper pools which
agrees with the depths of the unconformity in core data. In the absence of drill core and
particle size distribution data, we suggest that the elevation of pool depths measured soon
after a flood event can be used to approximate the base of the BPA. This will only provide a
minimum depth of the BPA base since the river is expected to deposit some sediment in
scoured areas during the flow recession. The thickness of a river's BPA is likely to depend on
the inter-relationship between several factors, including contemporary braidplain width,
sediment characteristics and the balance of sediment supply, the frequency and magnitude
of peak flow events, and the use of "hard engineering" to control a river's position. While
some of these factors are natural, the factors related to width and depth are influenced by
river engineering applied at each river. The Wairau River has the thickest BPA and has the
highest hydrological energy of the reaches studied. This river is considered by some river
engineers to be excessively narrowed, and a wider contemporary braidplain may result in a
dispersion of energy during flood flows, and subsequent thinning of the BPA. Interestingly,
the Ngaruroro River is also subject to high peak flows, although bed mobilising flows occur
less frequently than the Wairau. It is likely that the Ngaruroro BPA is being thinned by the
large volume of gravel extraction occurring within the study reach.

The LiDAR and bathymetric data gathered from our three study sites indicate that individual
river channels locally merge with and diverge from the water table surface of the BPA.
Water exchange across the bed is determined by bathymetry and hydraulic properties,
where the river can be forced above the water table by gravel lobes or dropped into the
water table in locations of scouring. Areas of relatively still water (pools) are therefore the
surface expression of the BPA, with seeps representing locations where the bedform drops
below the level of the BPA water table. This explains the occurrence of higher radon-222
activities in pools compared to flowing river channels, as well as differences in seasonal
water temperature.

For the interpretation of field data, the proposed conceptualisation highlights the
importance of identifying and knowing the nature of the relationship between these three
potential water sources to interpret observations. For example, to understand the nature
and magnitude of river-aquifer interaction by only sampling radon-222 in river channels can
be misleading. This is because the radon activity measured in the river depends both on the
river setting (run or pool), and the nature of the interaction between the BPA and regional
aquifer.

From a modelling perspective, it is questionable if a streambed conductance term is an
appropriate physical mechanism for representing braided river-aquifer exchange at the local
or catchment scale. The role of bed conductance, if significant, is to regulate exchange
between individual river channels and the BPA. Due to the relatively high transmissivity of
the bed materials, hyporheic exchange is an integral process of braided river flow, and
water can be seen to freely exchange between the surface and the bed. It follows that
consideration of water storage in the bed sediments (BPA) with a conductance term to
impede flow beneath those sediments would be a more appropriate approach for
simulating braided rivers at larger scales than to simulate individual channels with bed
conductance.

The BPA concept resolves vague definitions of "groundwater" in groundwater – surface
water interactions in braided rivers by considering the river as a whole system with an
associated subsurface storage component distinct from the regional groundwater system.
This provides specificity to the concept of a "river corridor" (Harvey and Gooseff, 2015),
where a river comprises not just the river channels, but the surrounding, fluvial deposits,
riparian zones, and floodplains between which hydrologic exchange occurs.

Braided river systems are spatially and temporally variable, which introduces heterogeneity
both within a BPA, and the adjacent older sediments. This heterogeneity can manifest as
preferential flowpaths, which can influence exchange fluxes at a local scale, as evident in
spatial variability of temperature and radon data. While the BPA consists of high
transmissivity sediments, and can itself be considered a preferential flow path at the
regional scale, the presence of preferential flow within the BPA at the local scale is not
captured by the conceptualisation presented here. We therefore recommend application of
the BPA concept at the regional scale, and to provide a hydrogeological context for local
scale studies. An additional consideration for applying the BPA concept is the volume of
reworked material associated with the river. In braided river environments, the volume of
gravel associated with the BPA is large, and significantly greater than the wetted channel
volume at average flow conditions. However, in some gravel bed rivers, the volume of
sediment mobilised by flooding events could potentially be very thin, and the relevance of
these mobile sediments on the exchange between the river and regional groundwater
system will depend on the scale of the study.
At the regional scale, the BPA concept is best applied to braided rivers that have stable or
actively degrading beds, or have had some form of bank stabilisation, which is common
where flooding is considered a risk to adjacent land. Stabilisation of river margins serves to
increase the frequency of gravel remobilisation within the active braidplain and prevent
reworking of adjacent bed material that may be part of the historical braidplain. Fine
sediment can percolate through the gravels or be deposited on the surface, and if these
gravels are not subsequently reworked, this can gradually consolidate and potentially clog
the pore spaces, accentuating the difference between contemporary and adjacent historical
braidplain sediments. Narrowing the area of braidplain that is reworked can thereby narrow
the BPA. The conceptualisation may be less appropriate, or the BPA boundaries may be less
distinct, where the contemporary braidplain margins are much wider than the active
braidplain and reworked over longer timeframes. This type of braided river behaviour is
typically seen in mountainous areas with low land use pressure, for example in the Southern
Alps of New Zealand, but would have occurred on lowland plains prior to widespread
engineering of river margins in New Zealand during the 1960s. If river channels can adjust
laterally over a wider area, it is expected that sediments within the contemporary braidplain
will be more heterogeneous, with channels of permeable recently mobilised gravel
intermingled with islands/areas of varying age and permeability. In these cases, the extent
of the BPA beyond the active braidplain across the contemporary braidplain may depend on
the frequency of contemporary braidplain reworking, or the connectivity with the active
braidplain. Similarly, it would be difficult to detect the base of the BPA in situations where
the river bed is rapidly aggrading.

**8.0 Conclusions**

By investigating the surface and subsurface sediment and saturation in three study reaches, we have developed a conceptualisation of how braided rivers exchange water at the local (reach) and sub-catchment (aquifer) scale. The interaction between the river system and groundwater can be considered to occur between (a) individual river channels and the BPA (hyporheic and parafluvial exchange), or (b) the braidplain and regional aquifers. Central to this conceptualisation is the presence of a braidplain aquifer (BPA), a thin (2-5 m) layer of loose, poorly sorted gravel which is formed via the process of bed mobilisation during flood-flows. The base of the BPA can be identified in drill core as an unconformity between poorly sorted unconsolidated gravels overlying more consolidated very poorly sorted gravels. Individual river channels can be hydraulically connected, transitional, or disconnected from the BPA, depending on the relationship between the braidplain water table and bathymetry. The nature of the hydraulic relationship between the braidplain and regional aquifers can also be hydraulically connected, transitional, or disconnected, depending on the relationship between the regional water table and base of the BPA gravels. Approaching the braided river as a (whole) system (river and BPA combined) enables field data to be interpreted within the context of its water source.

From a modelling perspective, the conceptualisation enables rivers to be appropriately represented at a local scale (river-BPA), or a regional scale (BPA/river system-regional aquifer), depending on the modelling objective. A parallel investigation not presented here is focussed on the implementation of the proposed conceptualisation into sub-catchment

scale models to quantify how changes in river morphology and BPA storage influence
recharge to the regional groundwater system. A key research gap is to understand the
relationship between BPA dimensions, bathymetry, and river flow dynamics, and this is a
subject for future research.

**Acknowledgements**
The authors would like to thank the New Zealand Ministry of Business, Innovation and
Employment for funding this research through the project "Subsurface processes in braided
rivers - hyporheic exchange and leakage to groundwater". We are also grateful to additional
support from Hawkes Bay Regional Council, Marlborough District Council and Environment
Canterbury. We also acknowledge the contribution of the Hydrogeophysics Group, Aarhus
University who carried out the EM surveys on our rivers, and Christy Songola who did the
radon sampling and analysis at our Selwyn study site. Lastly, we would like to express our
gratitude to the reviewers for their valuable comments, which helped us improve the
quality of this article.

**Financial support**
This research has been supported by the Ministry of Business, Innovation and Employment
(grant no. LVLX1901).

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
