# Peer review of "Conceptualising surface water-groundwater exchange in braided"

_EGUsphere, 2023_

## Author Comment (AC2)

**Review 1**

Braided rivers can provide substantial recharge to regional aquifers, with flow exchange between surface water and groundwater occurring at a range of spatial and temporal scales. However, the difficulty of measuring and modelling these complex and dynamic river systems has hampered process understanding and the upscaling necessary to quantify these fluxes. This is due to an incomplete understanding of the hydrogeological structures which control river-groundwater exchange. This paper presents a new conceptualization of subsurface processes in braided rivers based on observations of the main losing reaches of three braided rivers in New Zealand. It is useful to understand the braided river systems.

Dear Reviewer

Thank you for your review of our article. We understand that we should improve some aspects of this article and are willing to address your comments in a revised version. Please find below answers to your questions and explanations on how we would address your comments. Our responses in blue refer to lines in the original document, and your comments are in black.

Scott Wilson (on behalf of the co-authors)

**Major issues:**

1. Figure 9 in page 22, please explain that why the "glacial outwash gravels" are unsaturated?

   There is an error on the text in Fig. 9 which should say "variably saturated". We've corrected this, and have also modified Fig 9. to clarify the saturation status:

[Figure]

**Figure 9**. Subsurface resistivity collected by Electrical Resistivity Tomography (ERT) across Wairau and Waikirikiri river braids, looking downstream. Surfaces interpreted from the resistivity data are shown in black. An interpolated surface for the base of the BPA derived from drilling and bed data is shown in red.

The requirements for a variably saturated zone to form in the sediments underlying the BPA are described in the paragraph at line 172. For clarification, we will include the following explanation in the text at line 342:

Our explanation for the presence of this variably saturated zone is that the glacial outwash gravels are stratified, and vertical infiltration to the regional aquifer is limited by the lower permeability horizons of silt and clay. The presence of higher permeability horizons within the stratified postglacial sequence allows water to move laterally away from the recharge zone at a rate that exceeds vertical infiltration, enabling unsaturated or variably saturated conditions to form.

And how about the braidplain aquifer? Saturated or unsaturated?

We've added a sentence at line 154 to clarify the BPA saturation status:

For a perennially flowing river, the BPA will retain some degree of saturation, although unsaturated conditions may occur in the case of intermittent or ephemeral rivers if there are prolonged periods with no river flow.

Can you compare the ERT results with your borehole surveys in Fig. 4.

The contact at the base of the BPA shown on Fig 9 represents the unconformity revealed in drill core e.g. on Fig 4, and is described in section 5.1, including the grainsize distribution of the two units (Fig 5). While the locations of the ERT profiles don't exactly match the coring locations (Fig 3), there is structural and lateral consistency in both the coring and ERT data. We've added an interpolated surface to Figure 9 (see red line in the figure above), which has been derived by kriging intercepts of BPA base within the contemporary braidplain (in Fig 3). The sentence at line 457 has been modified for clarity:

These profiles reveal the contact between the braidplain aquifer and underlying sediments (the unconformities shown in Fig. 4) at elevations consistent with drillhole coring. For comparison, the elevation of the same unconformity, derived by kriging intercepts of the BPA base within the contemporary braidplain (Figure 3.), is shown in red on Figure 9.

2. In Fig. 10 at page 23, can you explain that what is the "DTS" and "A-DTS"?

These acronyms are explained in 4.1 (lines 298-299), and are also evident in the section 5.5 title. If the editor prefers, we can change the caption to:

Figure 10. Vertical (a) distributed temperature sensing (DTS)and (b) active-distributed temperature sensing (A-DTS) surveys carried out on the south bank of the Wairau River on w27. The river temperature over the period of the surveys is shown in (c) along with temperatures measured by DTS at 4 and 9 m depth. SWL=static water level measurements over the survey period

**Minor issues:**

These issues are specific to correct use of language. We agree with some of these changes, although we do disagree with some of them:

1. Line 17:"including:" should be changed to "including". changed
2. Line 26:"hyporheic" should be changed to "the hyporheic". not changed
3. Line 27:"exchange" should be changed to "the exchange". not changed
4. Line 28:"Exchange" should be changed to "The exchange". not changed
5. Line 30:"aquifer" should be changed to "aquifers". not changed
6. Line 55:"scale" should be changed to "scales". not changed

7. Line 95:"net" should be changed to "a net". changed

8. Line 100:"reach" should be changed to "reaches". not changed

9. Line 147:"recognises" should be changed to "recognizes". not changed

10. Line 190:"result" should be changed to "results". not changed

11. Line 218:"enables" should be changed to "enable". changed

12. Line 282:"are" should be changed to "were". changed

13. Line 291:"underling" should be changed to "underlying". changed

14. Line 301:"on" should be changed to "in". changed

15. Line 304:"insufficient" should be changed to "an insufficient". not changed

16. Line 331:"on Fig. 3" should be changed to "in Fig. 3". changed

17. Line 351:"on Fig. 4" should be changed to "in Fig. 4". changed

18. Line 351:"on Fig. 10" should be changed to "in Fig. 10". changed

19. Line 415:"are approximately" should be changed to "is approximately". not changed

20. Line 416:"condition" should be changed to "conditions". not changed

21. Line 441:"notable" should be changed to "a notable". changed

22. Line 446:"in the near surface" should be changed to "at the near surface". Changed to " in the near-surface, which improved"

23. Line 474:"in the 1960's" should be changed to "in the 1960s". changed

24. Line 501:"on Fig. 1" should be changed to "in Fig. 1". changed

25. Line 515:"losing water" should be changed to "lose water". changed to "be losing water"

26. Line 517:"therefore interpretation" should be changed to " therefore an interpretation". not changed

27. Line 550:"hyporheic exchange" should be changed to "the hyporheic exchange". not changed

---

## Author Comment (AC3)

**Review 2**

In this study, the authors present a revised conceptualisation of the sedimentary structure of braided river-aquifer systems as it is relevant for the quantification, upscaling and modelling of surface water-groundwater exchange fluxes on larger (regional to catchment) scales. The revised conceptualization builds on a range of different types of measurements, namely lidar, bathymetry, coring, particle size distribution, groundwater levels, water temperatures, radon-222 concentrations, electrical resistivity tomography and fiber optic distributed temperature sensing. The study is based on the analysis of three losing braided river reaches in New Zealand.

The authors demonstrate that the still active braidplain sections of a braided river system contain (or form) better sorted, largely unconsolidated sediments that can create a more conductive aquifer structure (which they call "braidplain aquifer"), while the formerly active, now inactive river sections develop more consolidated and clogged riverbeds or sediment layers due to the repeated deposition of fines and silty material during overbank flows. These "regional aquifers" are located underneath or adjacent to the braidplain aquifer. While these are essentially not new concepts, and riverbeds as separate hydrogeological units have long been considered are as the critical hydrogeological unit controlling exchanges in river-aquifer systems, the confirmation and extension of this concept to braided-river systems and a "braidplain aquifer" concept is still very valuable. What thus can be considered new here is the definition and characterization of an intermediate storage reservoir/aquifer layer which essentially extends the concept of riverbeds that act as modulating layers for the exchange between a (braided) river and an underlying aquifer to a system of larger vertical and horizontal extent and more spatial complexity. I believe that this concept is very helpful to better characterize, quantify and simulate the interactions between streams and aquifers in braided river systems, as it is, in principle, better suited to also address the highly preferential flow structures which one finds in such systems. These flow structures become slightly less important or more focused in deeper, older layers (buried paleochannels), but in the younger, shallow layers which lie close to the very active surface, the interactions and flow paths can be very complex and critically control exchanges not only of water but also of heat and mass.

Overall, the manuscript was nice to read. It is very clearly written and very well structured. I also really appreciated the clear definitions of the main terms in the introduction. It made the reading and understanding of the rest of the manuscript straightforward. Content wise, the updated conceptualization of a braided river-aquifer system can be adapted to many riverine environments and has a wide scientific relevance. I find that the following sentence of the discussion section nicely summarizes the applicability of the concept: "By identifying the base and margins of the BPA, and the process which forms it (reworking of bed material), the vertical and lateral extents to which hyporheic and parafluvial exchange occur can be identified by a change in sediment characteristics." In essence, the study shows that already relatively few key measurements are enough to identify the extent of the main layers as presented in their revised conceptualization, and that this knowledge can then be used to estimate water fluxes over much larger areas in these otherwise very complex braided river systems.

However, I also have some major points which I find have to be addressed first before the manuscript can be accepted. These as well as specific or minor points are listed in detail below. A major conceptual shortcoming is the fact that this intermediate aquifer layer between the river and the aquifer, i.e., what the authors call braidplain aquifer, is normally highly complex and can form highly conductive subsurface channels for preferential flow that, in case they are at some point covered by new sediments and not reworked anymore, can lead to buried paleochannels. The extended riverbed or "braidplain aquifer" concept therefore is primarily useful for simplified analyses on a very large scale, i.e., regional to catchment scale. On the reach or local scale, which was studied here, heterogeneity in the braidplain aquifer becomes extremely important for river-aquifer exchange fluxes, radon/residence time distributions, and flow paths in general. I think it is great to have a simplified conceptualization for regional scale assessments as it is presented in this generally excellent manuscript, but I miss an honest discussion of where this concept is NOT suited. Heterogeneity in these sediments is mentioned in the introduction, but the discussion section is void of any critical reflection of the new concept. I would highly appreciate some honest discussion about where this concept is usefull, and where it is too simplistic, as this will be very important information for practical implementations of the concept.

I also must point out that the radon measurements and analyses are not sufficiently well described or their uncertainty properly addressed. The residence times derived from the radon measurements are fantasies, as the employed method doesn't allow measurements precise enough to derive residence times beyond the 3-half-lives mark of Rn-222. But as the uncertainties of the measurements are not presented, the reader is not able to see this easily.

Given these shortcomings in an otherwise very nice manuscript, I recommend minor revisions before accepting the manuscript for final publication.

Dear Reviewer

Thank you for your comprehensive and constructive review of our article. We understand that we should improve some aspects of this article and are willing to address your comments in a revised version. Please find below answers to your questions and explanations on how we would address your comments. Our responses in blue refer to lines in the original document, and your comments are in black.

In response to your request for an honest discussion of where this concept is NOT suited, we refer you to lines 558-573 of the discussion, where we have suggested settings where the concept may not apply. However, we acknowledge that some additional discussion around heterogeneity is required here and will suggest an improvement to the discussion at the end of this document.

Scott Wilson (on behalf of the co-authors)

**Major comments**

Radon: Unfortunately, the entire radon aspect needs a thorough revision. The method is not sufficiently well described to allow an assessment of the validity/quality/uncertainty of the data (e.g., which water amount was analysed per sample?

250ml samples were analysed as per data collection section (refer to line 294).

How long were samples stored before being measured?

Samples were analysed 21-100 hours after sampling for Waikirikiri, and 20-24 hours for the Wairau.

How long were the samples measured for,i.e., how many counting cycles were employed?). ?)

To reduce the uncertainty of the WAT250 method using the Rad7, we increased the aeration time to 10 minutes, and the analysis duration recommended in the Durridge manual to 5 cycles of 10 minutes.

We have modified the relevant paragraph in the methods section as follows from line 273:

Piezometers were installed at different depths to provide a time series of water levels and temperature, and to enable sampling for radon-222 analysis. Samples for radon-222 analysis were collected in 250 ml bottles in the Waikirikiri (Songola 2022) and Wairau reaches from riffles, at different depths in pools, and from purged piezometers. Samples were analysed 21-100 hours after sampling for Waikirikiri, and 20-24 hours for the Wairau. Laboratory analysis for radon activity was conducted with a RAD7 (Durridge 2020a), RAD H2O (Durridge 2020b), and active DRYSTIK (Durridge 2021) in a closed loop system, with the results adjusted for decay since the time of sampling (WAT250 method). Additional radon measurements were made in the field

using the RAD AQUA method (Durridge 2020c) to verify the WAT250 method results, and these returned similar values. For this study we have reported the WAT250 data, which has a larger 2σ uncertainty associated with the measurements but enables more samples to be collected in a short time frame from remote field sites. To reduce the uncertainty of the WAT250 results, we increased the aeration time to 10 minutes, and the analysis duration recommended in the Durridge manual to 5 cycles of 10 minutes.

Durridge (2020c). Continuous Radon in Water Accessory for the RAD7 user manual.

Moreover, the (instrument reported) measurement uncertainties are not provided, nor is any assessment of the uncertainty of the radon measurements made.

We have reviewed the radon data and revised the values in Table 3, including 2σ uncertainties:

Table 3. Measured radon-222 activities (Bq m$^{-3}$) for the Wairau and Waikirikiri study reaches.

| River | Wairau | | | Waikirikiri | | | |
|---|---|---|---|---|---|---|---|
| Sample source | River run | Braidplain aquifer | Regional aquifer | River run | Braidplain aquifer | Variably saturated glacial outwash sediments | Regional aquifer |
| Samples | 16 | 13 | 21 | 10 | 38 | 7 | 6 |
| Rn min | 211 ± 108 | 322 ± 144 | 647 ± 202 | 200 ± 130 | 1490 ± 334 | 9180 ± 875 | 14150 ± 970 |
| Rn max | 525 ± 182 | 2513 ± 393 | 5008 ± 574 | 570 ± 190 | 7830 ± 720 | 12810 ± 1238 | 19970 ± 1165 |
| Rn mean | 345 ± 148 | 1131 ± 256 | 2880 ± 395 | 383 ± 166 | 4004 ± 517 | 10340 ± 853 | 16842 ± 1045 |

The residence times that are calculated based on the measured radon activities are fantasies, as the precision of the employed method is not good enough (it is impossible to use the Rad H2O method and get radon concentration measurements at a precision that would allow a residence time detection beyond around the 3-half-lives mark of radon (ie. about 12 days), and this without even considering the fact that most likely the samples were not measured directly but first stored for a day or two, leading to a further loss in precision).

Thank you for this comment which we agree with. We have accordingly included the measurement uncertainties and removed residence times from Table 3. To clarify, the WAT250 method results were adjusted for decay since the time of sampling. The decay correction factor (DCF) is given by the formula DCF = exp(T/132.4), where T is the decay time in hours (Durridge Rad H2O manual).

Last but not least, the secular equilibrium for one of the two measured sites is in my opinion not chosen appropriately, as the regional aquifer shows much more elevated background activity concentrations. I don't see why the braidplain aquifer should be used to define the background values instead of the regional aquifer, unless the sediments are very very different, which is very unlikely. Even though this aspect of the study needs a major overhaul, looking at the data tells me that the main conceptual conclusions from the data will most likely remain valid.

We see no reason to assume that two different gravel deposits would have similar secular emanation values, or vice versa. We have chosen our equilibrium values on the best available information but agree that some caution needs to be advised in the text. We will address this in your specific comments.

Lidar-Bathymetry data: Although mentioned as the first two data types in the abstract, this data was not presented in the paper. It is only used to discuss the concept and derive conclusions. Since this data is important for the assessment of braided river-aquifer systems, and critical for the derivation of the updated conceptualization of braided river-aquifer systems in this manuscript, it must be provided somehow.

We agree, and not including any of our bathymetry or lidar is an oversight. Displaying the actual bathy/LiDAR data is perhaps not particularly informative, since it's the specific querying of the bathy-lidar that provides useful information, not the bathy-lidar surfaces per se. With this in mind, we have created an additional figure

(Fig 4), which shows two maps for the Wairau at same position and scale to illustrate the relationship between the river surface, river bed, and kriged BPA water table: (a) the difference between the river surface and static water level with aerial imagery as backdrop (b) the difference between the static water level and river bed. An additional paragraph has been added to the filed data section at line 273:

LiDAR data was captured in dry areas of riverbed using a LiDARUSA Snoopy LiDAR scanner deployed on either a UAV or backpack. Bathymetry and water surface elevation were mapped using a kayak or remote controlled jetboat equipped with a paired RTK GPS and echosounder, and wading with an RTK GPS. Interpolation, or (where necessary) optical-bathymetry techniques, were used generate high-resolution bathymetry maps from less-dense echosounder survey data. The dry topography from LiDAR was stitched together with the bathymetry data to provide a complete digital elevation model (DEM) for each reach at a spatial resolution of 1 m or less, and a vertical accuracy of ±0.1 m in dry areas and ±0.2 m in wet areas.

A new section (5.2) has been written to describe the data shown in the new Figure 4:

5.2 LiDAR and bathymetry

LiDAR and bathymetry surveys were carried out in each study area to understand the spatially varying relationship between the river surface, bed levels, and water levels in the braidplain and regional aquifers. Repeat surveys were carried out following significant flood events to capture changes in bed levels. An example of our LiDAR and bed elevation data for the Wairau River is shown in Figure 4. These data were captured on 19 Feb 2020 at relatively low flow conditions, measured at 13.4 $m^3.s^{-1}$ to 11.5 $m^3.s^{-1}$ (±3 %) at the upstream and downstream margins of the image, respectively. The river water surface and bed elevation data within the wetted channel are shown on Figure 4 in relation to a modelled surface of hydraulic head across the river system, represented by piezometric contours which are shown in Figure 4a. This surface was fitted (sum squared error of 5 x $10^{-6}$) to 25 water level observations (yellow points in Fig 4a) located within and outside of the contemporary braidplain by universal kriging with an exponential variogram of anisotropy of 0.9 at 090°, partial sill 0.31 m, and range 670 m. With such a large variogram range, the surface should be considered as indicative of an averaged hydraulic head across the regional and braidplain aquifers. The kriged surface does reveal an inflection of the piezometric contours across the contemporary braidplain margins, indicating that flow within the BPA is largely controlled by river exchange and preferential flow within the BPA, with flow being approximately sub-parallel to the contemporary braidplain longitudinal orientation.

Fig. 4a reveals locations in the river system where the river water surface is higher than the braidplain water table (red and orange zones), indicating that the river is losing flow to the BPA in these areas. Areas of the river which are coloured blue in Fig. 4a represent the surface expression of the braidplain water table in pools. These are locations where the river can potentially gain flow. The black areas denoted as "riffles" are identified from a slope raster derived from the digital elevation model (DEM). Locations where maximum potential river water loss occurs can be identified in most cases as being situated at the upstream margins of high elevation riffles.

The bathymetry DEM (Fig. 4b) reveals the presence of scouring along the contemporary braidplain margins, which in the case of the Wairau River is promoted by excessive river narrowing and rock training banks. The corollary of this scouring is the relative mounding of gravel in the middle of the contemporary braidplain. The difference between the river bed level and hydraulic head reveals locations where the river bed is above the braidplain aquifer, and has the potential to be losing-disconnected at these locations. In most cases these areas also correspond to the upstream margins of high elevation riffles.

[Figure]

**Figure 4. Images of the wetted Wairau River channel showing differences between (a) river water surface and a kriged hydraulic head (overlain on aerial imagery), and (b) bathymetry and kriged hydraulic head (overlain on the DEM). River flow is from left to right.**

**Specific comments:**

L 103-106: a more recent modelling approach for braided river-aquifer systems with their buried paleochannels that should be referred to here is Schilling, O. S., Partington, D. J., Doherty, J., Kipfer, R., Hunkeler, D., & Brunner, P. (2022). Buried paleo-channel detection with a groundwater model, tracer-based observations, and spatially varying, preferred anisotropy pilot point calibration. Geophys. Res. Lett., 49(14), e2022GL098944. https://doi.org/10.1029/2022GL098944

*We have now included this reference in the references list and have cited it in the text at line 105:*

*A significant body of literature exists to describe braided river deposits via morphology (Huber and Huggenberger 2015), sedimentology (Huggenberger and Regli 2006, Theel et al. 2020), geophysics (Pirot et al. 2019), and modelling approaches (Pirot et al. 2014; 2015, Brunner et al. 2017, Schilling et al. 2022).*

L 107-109: Indeed, changing bed morphologies create problems for the representation of braided river systems (and this problem isn't even restricted to braided river systems but occurs also in less complex river

corridors). One way to deal with this problem when one simulates SW-GW interactions in braided river systems is using data assimilation, where whenever new information on the bathymetry of the river system becomes available, that information gets incorporated into the forward model. The concept was demonstrated by Tang, Q., Schilling, O. S., Kurtz, W., Brunner, P., Vereecken, H., & Hendricks Franssen, H.-J. (2018). Simulating flood induced riverbed transience using unmanned aerial vehicles, physically-based hydrological modelling and the ensemble Kalman filter. Water Resour. Res. https://doi.org/10.1029/2018WR023067. Although not necessarily feasible or required on the regional scale, this way forward should nevertheless be mentioned here. Another approach that can be useful relies not on assimilation of riverbed bathymetry information but on a moving pilot points approach, which can be used to "follow" changing structures simply via inversion of hydraulic- and tracer-based data: Khambhammettu, P., Renard, P., & Doherty, J. (2020). The traveling pilot point method. A novel approach to parameterize the inverse problem for categorical fields. Adv. Water Resour., 138, 103556. https://doi.org/10.1016/j.advwatres.2020.103556. Also this study should be mentioned here.

This comment looks to be referring to lines 117-119. Thank you for pointing these two papers out, they are superb developments. We have changed the last para (lines 118-119) to:

In recent years, two approaches to simulate the transitions of dynamic bed morphology and sediments on river-groundwater exchanges have been tested. The first approach applied the ensemble Kalman filter and areal imagery to assimilate river bed topography and to update aquifer hydraulic conductivities in a HydroGeoSphere model for a 2-km section of the Emme River in Switzerland (Tang et al. 2018). The data assimilation scheme strongly improved predictions of post-flood hydraulic states of the system. The second approach proposed a pilot point parametrization scheme where both the aquifer properties (hydraulic conductivity) and the location of the pilot points are inferred, e.g. from river-bed training images (Khambhammettu et al. 2020). The corresponding Traveling Pilot points (TRIPS) scheme could potentially be used to describe the transition between discrete states of river morphology. To some extent these approaches enable the application of fully integrated 3D models in dynamic river environments of appropriate scale, although their application in a larger river system or at a larger scale is untested.

l192-211: It is worth mentioning here that this sedimentological and SW-GW interaction concept of braided, alluvial gravel rivers was already presented a rather long time ago, and moreover gave the different types of gravel-sand sediments a full nomenclature. The key reference to mention here is: Huggenberger, P., Hoehn, E., Beschta, R., & Woessner, W. (1998). Abiotic aspects of channels and floodplains in riparian ecology. Freshwater Biol., 40, 407-425. https://doi.org/10.1046/j.1365-2427.1998.00371.x

We have included this reference, and an additional sentence to accommodate this at line 208:

The sedimentological features and groundwater-surface water interaction concepts associated within the contemporary braidplain have been identified and detailed by previous authors (e.g., Huggenberger et al. 1998). Regardless of the nature of the relationship between the braidplain and regional aquifers, the braidplain gravels have a higher transmissivity than both the adjacent and underlying sediments, because of repeated reworking of the braidplain gravels during high flow events…..

l226: don't superscript the "s" in the unit

This seems to be an error generated by the pdf conversion. The "s" is not superscript in the original document.

l228-334: Unfortunately, the findings that the electromagnetic methods do not help in identifying the structure of braided river sediments is not surprising and only confirms what could be observed previously. I disagree however that the magnitude of the resistivity is the main cause for these methods to not work. The main reason is rather that the resistivity of the sediments are too similar, as they are in fact all more or less the same type of gravel and sand, simply differing in whether they are more washed or aligned in different ways. In order to be able to differentiate these types of sediment structures one has to resort to ground penetrating radar. The effectiveness of the method has, for example, been demonstrated in the publication I mentioned above: Huggenberger, P., Hoehn, E., Beschta, R., & Woessner, W. (1998). Abiotic aspects of channels and

floodplains in riparian ecology. Freshwater Biol., 40, 407-425. https://doi.org/10.1046/j.1365-2427.1998.00371.x

or in more detail in the following publication: Huggenberger, P., Meier, E., & Pugin, A. (1994). Ground-probing radar as a tool for heterogeneity estimation in gravel deposits: advances in data processing and facies analysis. J. Appl. Geophys., 31(1-4), 171-184. https://doi.org/10.1016/0926-9851(94)90056-6

I would prefer it if the author's revise the statement on why EM methods do/did not work in their case and highlight somewhere that ground penetrating radar could be a way forward in this respect.

Thank you for raising this point. When we began this investigation, we didn't know which geophysics methods would be best applied in our braided rivers, so were open minded. The first method we did test was GPR (50, 100, 200 and 500 MHz) on 5 transects in the active and contemporary Waikirikiri braidplain. We neglected to include GPR in the field methods section because the results were inconclusive. The GPR surveys revealed the water table surface but failed to reveal any clear underlying structure corresponding with our drill core at this site. Our interpretation of the poor performance of GPR is two-fold. Firstly, water has a high dialectric constant, so most of the GPR signal is reflected at the water table. Secondly, beneath the water table, the signal is rapidly attenuated with depth by reflection and dispersion due to the presence of cobbles and boulders, which are abundant at the Waikirikiri field site. As an explanation for this, a feature >= 1/10 of the wavelength will increasingly scatter the signal, so a 100 MHz signal will be scattered by 0.3 m features (boulders) and a 200 MHz signal will be scattered by 0.15 m features (cobbles). It seems likely that GPR would be successful in detecting subsurface features in settings with less cobble and boulders, which is not the case in the three field sites we have studied. The best of the GPR profiles we captured is shown below:

[Figure]

The near-surface sediments at our study sites are extremely resistive > 2000 ohm-m (the scale on Fig 9 has been maximised at 4000 ohm-m, as have the values on the profile). It is worth noting that even the river water is resistive (fluid specific conductance ~5 mS/m). Our experience is that we could identify the base of the BPA in ERT surveys at an elevation consistent with drill core elevations, but only in surveys carried out across the wetted river channel. The most likely explanation for this is the increased conductivity caused by the saturated sediment and filled pore space, which has improved the conductivity response. It's well known that electric conductance is also enhanced in the presence of fine material, particularly clays, which are likely to be present within the river channel. However, we agree that in most cases we are trying to detect differences between high and very resistivity, so our ability to identify clear structural features will be compromised.

Based on our experience we think it's best to trial a few geophysical methods and see what works. For braided rivers in New Zealand, we would recommend using GPR for mapping the water table, and trial ERT within wetted channels to investigate subsurface structure. The use of NMR in braided rivers looks to be quite promising. We have written a paper which looks at the application of NMR at these field sites, however this paper is currently under review, hence this method is not covered here.

We will add the 5 GPR surveys to Table 2, and revise the paragraph at line 451 to include on our experience with GPR, and revise the reason for poor resistivity response as follows:

Hydrogeophysical methods were also used to image the subsurface, including passive (DTS) and active (ADTS) distributed temperature sensing (Banks et al., 2022), ground penetrating radar (GPR), electrical resistivity tomography (ERT), transient electromagnetic (tTEM) and electromagnetic induction (DualEM421). The tracks prepared for tTEM and DualEM surveys are evident in the aerial photos shown in Figure 3. SkyTEM data were also available for the Ngaruroro area (Rawlinson et al. 2021). Of the hydrogeophysical methods employed, DTS/ADTS, and ERT were the most successful methods for delineating sediment structure and saturation associated with the river. The resistivity of New Zealand braided river water (fluid specific conductance ~5 mS/m) and associated gravel deposits is very high (400-10,000 Ωm). For this reason, we think there was insufficient resistivity contrast for electromagnetic and ERT methods to reveal distinct subsurface features in most of our surveys. SkyTEM data did provide good definition of the basement contact beneath the Ngaruroro River but did not reveal any clear structural features in the near surface (<10 m). GPR surveys that were trialled at the Waikirikiri site clearly revealed the shallow water table but did not reveal any clear structure beneath the water table due to reflection of the signal.

Table 2. Type and number of measurements undertaken in the three study reaches.

| Measurement type | Ngaruroro | Wairau | Waikirikiri |
|---|---|---|---|
| Differential flow gauging | 2 | 2 | 14 |
| Local river stage/temperature | | 3 | 6 |
| LiDAR and bathymetry surveys | 1 | 2 | 2 |
| Piezometers | 19 | 31 | 43 |
| Cored holes | 10 | 8 | 21 |
| Particle size distribution | 36 | 38 | 60 |
| Core porosity | 6 | 12 | 5 |
| Field porosity | 3 | 10 | 4 |
| Radon-222 samples | 5 | 53 | 61 |
| tTEM | Y | Y | Y |
| DualEM | Y | Y | Y |
| SkyTEM | Y | | |
| ERT surveys | | 9 | 11 |
| Ground penetrating radar (GPR) | | | 5 |
| DTS installations (vertical) | | 2 | 3 |
| DTS installations (horizontal) | | | 2 |

l 429, and following: unclear what 'BPA source' is supposed to be. Moreover, the following sentence is repeated twice.

We have revised the radon section, and in doing so have removed 'BPA source' from the text, and the repeated sentence.

l435-437: Looking at table 3, the regional aquifer shows much larger Rn-222 activities than the BPA. Why do the authors believe that it is justified to use the largest measured BPA Rn-222 activity concentrations as the secular equilibrium values? It makes much more sense to me to use the regional aquifer values, assuming that the regional aquifer also consists of sediments from the Waikirikiri river, albeit sediments that were deposited a longer time ago. Using the regional aquifer values as secular equilibrium indicators would result in much lower residence times for the BPA water in Waikirikiri. I believe that the RT of 15.5 days (essentially the upper detection limit of the RAD H2O based Rn-222 method) is too old and in reality would lie closer to the values observed for the Wairau AQ.

This is a really good point. We think the Wairau BPA activities are small because the residence time is not sufficient for samples to reach equilibrium. Accordingly, we have to resort to using the Wairau aquifer secular

equilibrium to represent an equilibrium value for the Wairau BPA (line 437). A possible reason for the large difference in the Wairau and Waikirikiri residence times is the differing relationship between the active and contemporary braidplains. In the Wairau system, the active and contemporary braidplains cover the same extent. By contrast the Waikiririkiri system has a very narrow active braidplain and considerably wider contemporary braidplain (see area of clean gravels vs contemporary braidplain on Fig 3). From a hydrological perspective this creates comparatively longer parafluvial pathways in the Waikirikiri braidplain aquifer. This may explain the larger Waikirikiri travel time estimates, although we also acknowledge that the secular equilibrium of both braidplain gravel units is not well constrained. Further studies at the site are conducting a large volume of batch experiments on the sedimentary material from the drill cores in an attempt to better constrain the radon activity at secular equilibrium.

To get an indication of the Waikirikiri BPA equilibrium, we did a stochastic comparison of travel times derived from temperature time series and radon derived residence times. This comparison showed that the temperature-derived travels times could not be matched using an equilibrium >10,000 BQ/m3, while values based on hydrogeological knowledge (7,500-9,000) compared well. However, we think that a temperature-radon comparison is beyond the scope of this paper, and we intend to write a paper on that aspect alone.

l437-438: What is the basis for the assumption that the secular EQ for the Wairau AQ is 4800 Bq/m3 ? For the Waikirikiri site it is explained how the secular EQ was derived, but for the Wairau aquifer it is not. From table 3 I understand that the 4800 Bq/m3 was the highest value measured in the regional aquifer, which makes sense to me. But please explain.

The Wairau aquifer equilibrium was derived by plotting measured radon activity vs the piezometer distance from the river. A curve could then be fitted to the data using the ingrowth equation[1] to give an estimate of the equilibrium. We found that both the BPA and regional aquifer samples fitted to the same curve. Previous samples reported by ESR[2] from the regional aquifer at sites downgradient of our field site in the Wairau returned radon activities around 7000 BQ/m$^3$. However, the ESR activities do not fall on the same ingrowth pathway as our samples, and we think those samples are sourced from sediments associated with an earlier depositional phase (early to mid-Holocene).

table 3: What are the measurement uncertainties of the Radon measurements? The RAD H2O method is notoriously uncertain if one follows the standard protocol of the manufacturer. Which steps were taken to reduce the measurement uncertainty of the standard measurement protocol? If the default procedure to measure Rad7 in the 250ml Rad H2O bottles was used, then it is highly likely that the instrument reported measurement uncertainties were an order of magnitude larger than the measured values shown for the river water samples.

We have updated Table 3 to include the 2σ uncertainties derived by the Durridge Capture Rad7 data acquisition software. To reduce the uncertainty of the WAT250 method results, we increased the aeration time to 10 minutes, and the analysis duration recommended in the Durridge manual to 5 cycles of 10 minutes.

In other words: In order for the reader to be able to assess the quality and validity of the presented radon-222 data with the specific method that the authors used, the authors must provide information on the measurement protocol that was employed and report the instrument-reported measurement uncertainties. This is especially important as the uncertainties in the activities have a huge effect on the uncertainty of the residence time estimates.

We have modified the data collection section from line 294 to be more specific about the measurement protocol and uncertainties.
* * *
[1] Close, M. 2014. Analysis of Radon data from the Wairau River and adjoining Wairau Plains Aquifer February 2014. Institute of Environmental Science and Research Client Report CSC14001 for Marlborough District Council.

[2] Hoehn, E., von Gunten, H.R. 1989: Radon in Groundwater: A Tool to Assess Infiltration From Surface Waters to Aquifers. Water Resources Research, 25(8): 1795-1803.

table 3: estimating a residence time of 25 days is not possible with the Rn-222 method, unless maybe one has an extremely precise radon-222 measurement system, for example a very well tuned liquid scintillation counter. With the Rad7 instrument, and particularly with the Rad H2O grab sampling technique, uncertainties are much too large to estimate Rn-222 based residence times beyond the 3 half-lives mark, i.e. beyond about 11.5 days.... The fact that the authors provide such values without a proper discussion of the employed method or the uncertainties and detection limits again tells me that a thorough re-thinking and revision of the radon-222 methodology is necessary.

Yes we agree with this statement, and accordingly have removed residence times from Table 3. We did compare RAD AQUA and WAT250 samples at both sites, more comparisons were made at the Waikrikiri site due to its closer proximity to our university. The two methods return very similar results for both sites with r2 0.99 for Wairau (n=7) and 0.95 for Waikirikiri (n=44), although the RAD AQUA uncertainties are about a third of WAT250 values because of the different sampling setup and method of the Rad7. The Waikirikiri data are reported in (Songola 2022), and a paper based on that work has recently been submitted to Journal of Hydrology. So, while the WAT250 measurement uncertainties are large, we can have confidence that the residence times are good estimates. What we suggest is a revised radon section (Section 5.3 from line 428) as follows:

A summary of the radon-222 results and measurement uncertainties for surface water and groundwater sources in the Wairau and Waikirikiri reaches is shown in Table 3. In the Wairau system, samples from riverbed piezometers and riverbed seepages have similar radon activities with ranges 355-2500 $BQ.m^{-3}$ and 320-1450 $BQ.m^{-3}$ respectively. Accordingly, samples from seepages and riverbed piezometers are both considered to represent the braidplain aquifer.

The radon data show distinct groupings, with radon-222 activities increasing from river channel to BPA to regional aquifer. At both sites, radon activities in river run samples were significantly lower than those in BPA samples. In the Wairau study reach, there is a notable overlap in radon activities between the braidplain and regional aquifers, indicating a likely hydraulic connection between these two systems. Conversely, in the Waikirikiri study reach, there is a downward increase in radon activities from the BPA to the variably saturated zone and further into the regional aquifer, with no overlapping values. This suggests a hydraulic disconnection between the BPA and the regional aquifer in the Waikirikiri reach.

Table 3

The determination of residence times between the river and each aquifer depends on knowing the initial channel condition, representative secular equilibrium for the host gravel deposit, as well as a well-defined flow path length. Our estimate of the initial river channel condition is 180 $BQ.m^{-3}$ for Wairau and 200 $BQ.m^{-3}$ for Waikirikiri, reflecting the lowest measured river radon-222 activities. A secular equilibrium estimate of 4800 $BQ.m^{-3}$ was derived for Wairau aquifer samples by plotting measured groundwater radon activity against distance of the piezometer from the river and fitting the ingrowth equation to the data to determine the 21 day equilibrium value. This exercise indicated that the Wairau BPA activities are too low for samples to reach equilibrium. In the absence of sediment specific data, the Wairau aquifer secular equilibrium was chosen to represent the Wairau BPA equilibrium. The secular equilibrium for the Waikirikiri BPA is estimated at 8000 BQ.m-3 based on the highest activity observed in the BPA (7450 $BQ.m^{-3}$). The lowest activity is observed in porewater samples from piezometer sumps in the variably saturated glacial outwash gravels beneath the braidplain aquifer (9180 $BQ.m^{-3}$). Based on the secular equilibrium values chosen, residence times for our study reach samples are estimated to be in the range of 0.2 to 5.5 days for the Wairau BPA in the study reach and 1 to >12 days for the Waikirikiri BPA. Due to the large uncertainties associated with the WAT250 method, these estimates should be considered for comparative purposes only.

L471: "The hydrogeologic structure..."

We've corrected the text to: "The hydrogeologic structure in the Wairau study reach…"

L535-540: Lidar and bathymetric data were not presented and therefore can't be used here to discuss aspects/the functioning of the system... or the other way around: if such data are used to derive insights/conceptualisations of braided river-aquifer systems functioning, the respective data must also be presented and discussed properly.

This paragraph is based on field observations of differences in elevation that are apparent on the ground and reinforced by elevation data. We've addressed this comment by including a new section (5.2) and new Figure 4.

**Discussion**

We suggest a revision to the discussion at line 558 (new text in blue) as follows:

Braided river systems are spatially and temporally variable, which introduces heterogeneity both within a BPA, and the adjacent older sediments. This heterogeneity can manifest as preferential flowpaths, which can greatly influence exchange fluxes at a local scale, as evident in spatial variability of temperature and radon data. While the BPA consists of high transmissivity sediments, and can itself be considered a preferential flow path at the regional scale, the presence of preferential flow within the BPA at the local scale is not captured by the conceptualisation presented here. We therefore recommend application of the BPA concept at the regional scale, and to provide a hydrogeological context for local scale studies. An additional consideration for applying the BPA concept is the volume of reworked material associated with the river. In braided river environments, the volume of gravel associated with the BPA is large, and significantly greater than the wetted channel volume at average flow conditions. However, in some gravel bed rivers, the volume of sediment mobilised by flooding events could potentially be very thin, and the relevance of these mobile sediments on the exchange between the river and regional groundwater system will depend on the scale of the study.

At the regional scale, the BPA concept is best applied to braided rivers that have stable or actively degrading beds, or have had some form of bank stabilisation…..